# Distinct subpopulations of mechanosensory chordotonal organ neurons elicit grooming of the fruit fly antennae

Stefanie Hampel[1†]*, Katharina Eichler[1†], Daichi Yamada[2], Davi D Bock[3], Azusa Kamikouchi[2], Andrew M Seeds[1]*

[1]Institute of Neurobiology, University of Puerto Rico Medical Sciences Campus, San Juan, Puerto Rico; [2]Division of Biological Science, Graduate School of Science, Nagoya University, Nagoya, Japan; [3]Department of Neurological Sciences, Larner College of Medicine, University of Vermont, Burlington, United States

*For correspondence:
stef.hampel@gmail.com (SH);
seeds.andrew@gmail.com (AMS)

†These authors contributed equally to this work

Competing interests: The authors declare that no competing interests exist.

**Abstract** Diverse mechanosensory neurons detect different mechanical forces that can impact animal behavior. Yet our understanding of the anatomical and physiological diversity of these neurons and the behaviors that they influence is limited. We previously discovered that grooming of the *Drosophila melanogaster* antennae is elicited by an antennal mechanosensory chordotonal organ, the Johnston's organ (JO) (Hampel et al., 2015). Here, we describe anatomically and physiologically distinct JO mechanosensory neuron subpopulations that each elicit antennal grooming. We show that the subpopulations project to different, discrete zones in the brain and differ in their responses to mechanical stimulation of the antennae. Although activation of each subpopulation elicits antennal grooming, distinct subpopulations also elicit the additional behaviors of wing flapping or backward locomotion. Our results provide a comprehensive description of the diversity of mechanosensory neurons in the JO, and reveal that distinct JO subpopulations can elicit both common and distinct behavioral responses.

## Introduction

Animals can detect complex mechanical forces in their environments through diverse mechanosensory neuron types that produce different sensations and influence appropriate behavioral responses. These neurons display diverse tuning to mechanical stimuli and differ widely in their peripheral and central nervous system (CNS) projections (*Abraira and Ginty, 2013*; *Tuthill and Wilson, 2016*). One defining feature of mechanosensory neurons is that their axonal projections from the body periphery terminate in an orderly topographical arrangement in discrete zones of the CNS. Different types of topographical organization are described across the diversity of mechanosensory neuron types based on features such as their responses to particular frequencies of sound (tonotopy) or their locations across the body (somatotopy) (*Appler and Goodrich, 2011*; *Erzurumlu et al., 2010*; *Muniak et al., 2015*). Although topographical organization is thought to provide a means by which sensory neurons connect with the appropriate neural circuits in the CNS that facilitate relevant behavioral responses (*Kaas, 1997*; *Thivierge and Marcus, 2007*), it remains unclear how sensory topography interfaces with the CNS behavioral circuitry. Critical for addressing this question is to define the anatomical and physiological diversity of the mechanosensory neurons that make up this topography, and link them to the diverse behaviors that they influence.

The *Drosophila melanogaster* Johnston's organ (JO), a chordotonal organ in the antennae, is an excellent system in which to study how mechanosensory topography influences behavior. The JO

detects diverse types of mechanical forces that move the antennae, including sound, wind, gravity, wing beats, and tactile displacements (*Hampel et al., 2015*; *Ishikawa et al., 2017*; *Kamikouchi et al., 2009*; *Mamiya and Dickinson, 2015*; *Matsuo et al., 2014*; *Patella and Wilson, 2018*; *Yorozu et al., 2009*). The ability of the JO to respond to these different stimuli is conferred by about 480 mechanosensory neurons called JONs (*Kamikouchi et al., 2006*). Subpopulations of JONs are selectively excited by different vibrational frequencies or by sustained displacements of the antennae and send their projections into discrete zones in the CNS (*Kamikouchi et al., 2006*). In accordance with their diverse physiological tuning properties, the JONs are implicated in controlling diverse behaviors including courtship, locomotion, gravitaxis, wind-guided orientation, escape, flight, and grooming (*Hampel et al., 2015*; *Kamikouchi et al., 2009*; *Lehnert et al., 2013*; *Mamiya et al., 2011*; *Mamiya and Dickinson, 2015*; *Suver et al., 2019*; *Tootoonian et al., 2012*; *Vaughan et al., 2014*; *Yorozu et al., 2009*). However, efforts to define how the different JONs interface with downstream neural circuitry to influence these behaviors have been hampered by the incomplete description of the morphologically heterogeneous JON types within each subpopulation (*Kamikouchi et al., 2006*; *Kim et al., 2020*). Furthermore, our understanding of the diversity of behaviors that are influenced by different JON subpopulations remains incomplete.

We previously discovered that activation of different JON subpopulations elicits antennal grooming behavior (*Hampel et al., 2015*), which involves the grasping and brushing of the antennae by the front legs (*Böröczky et al., 2013*; *Robinson, 1996*). However, we did not determine the extent to which these subpopulations were anatomically and physiologically distinct from each other, or whether behaviors other than grooming could be elicited by activating these JONs. In work presented here, we first define the JON morphological diversity by reconstructing major portions of each subpopulation from a complete serial-section electron microscopy (EM) volume of the adult fruit fly brain (*Zheng et al., 2018*). We next produce transgenic driver lines that selectively target expression in each subpopulation. These lines enable us to visualize the distribution of the different subpopulations in the antennae and determine that they respond differently to mechanical stimuli. Optogenetic activation experiments confirm our previous finding that each JON subpopulation can elicit grooming of the antennae (*Hampel et al., 2015*). However, we report here that one subpopulation of JONs also elicits wing flapping movements while another subpopulation elicits the avoidance response of backward locomotion. Collectively, our results provide a comprehensive description of the topography of the JO, and reveal that different JON subpopulations whose projections occupy different points in topographical space can elicit common and distinct behavioral responses.

## Results

### EM-based reconstruction of different JON subpopulations

We first sought to define the morphological diversity of the neurons within each JON subpopulation. JONs project from the antennae through the antennal nerve into a region of the ventral brain called the antennal mechanosensory and motor center (AMMC) (*Figure 1A*). The projections of different subpopulations form discrete zones in the AMMC (zones A-F, *Figure 1B*). JONs that respond exclusively to antennal vibrations project laterally into zones A and B (called JO-A and -B neurons), whereas JONs that are tuned to antennal vibrations and/or sustained displacements project medially into zones C-E (JO-C, -D, and -E neurons) (*Kamikouchi et al., 2006*; *Mamiya and Dickinson, 2015*; *Matsuo et al., 2014*; *Patella and Wilson, 2018*; *Yorozu et al., 2009*). We previously discovered the aJO subpopulation of JONs (*Hampel et al., 2015*), which we now rename as 'JO-F' neurons based on their projections to 'zone F' that we newly designate here. JO-F neurons enter the brain through the AMMC like other JONs, but then project ventrally (*Figure 1A,B*, blue JONs) (*Hampel et al., 2015*). While the majority of JONs project to a single zone, additional JONs have been described that have branches projecting to multiple zones (called JO-mz neurons) (*Kamikouchi et al., 2006*). Because our previous work implicated JO-C, -E, and -F neurons in antennal grooming behavior, we reconstructed these subpopulations within a serial-section EM volume of the entire fruit fly brain to define their morphological diversity (*Zheng et al., 2018*).

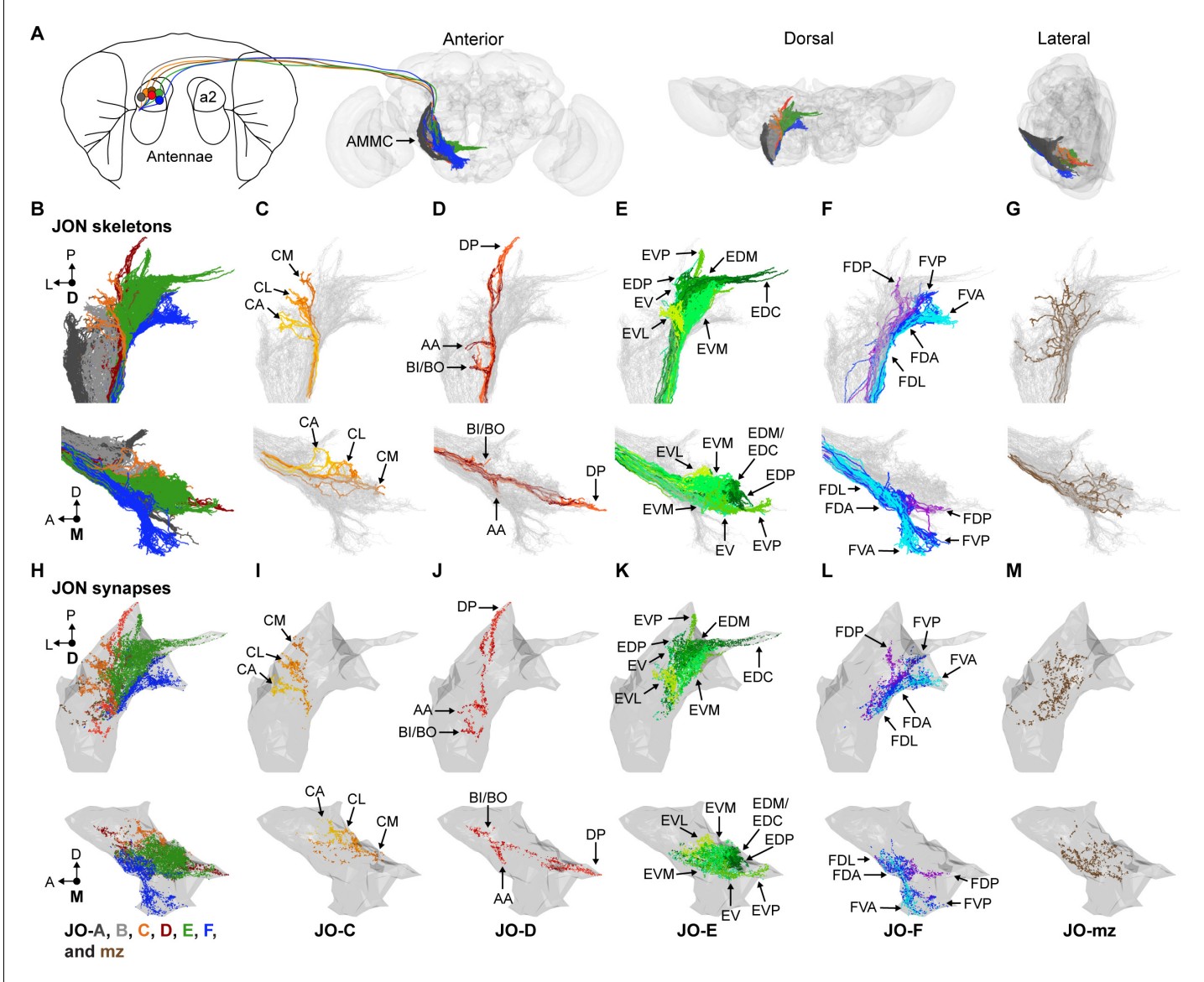

**Figure 1.** EM-based reconstruction of JONs. (**A**) JON projections from the second antennal segment (a2) into the AMMC brain region (brain neuropile shown in gray). Anterior, dorsal, and lateral views of reconstructed JONs are shown. (**B–G**) Reconstructed JONs are shown from dorsal (top) and medial (bottom) views. (**H–M**) Dorsal (top) and medial (bottom) views of the JON pre- and post-synaptic sites are shown (colored dots) with a gray mesh that outlines the entire reconstructed JON population. See *Figure 1—figure supplement 2* for pre- versus post-synaptic site distributions. All reconstructed JONs are shown in (**B–G**), but only fully reconstructed JONs are shown in (**H–M**) (JO-A and -B synapses not shown). JO-A and -B neurons were previously reconstructed by *Kim et al., 2020*. Colors in (**A**, **B**, and **H**) correspond to the zones to which the different JONs project, including zones A (dark gray), B (light gray), C (orange), D (red), E (green), F (blue), and mz (brown). Panels (**C–G**) and (**I–M**) show JONs that project specifically to zones C (**C,I**), D (**D,J**), E (**E,K**), F (**F,L**), or multiple zones (mz) (**G,M**). Color shades in (**C–G**) and (**I–M**) indicate different JON types that project to that zone. Zone subareas are indicated with labeled arrows. See *Video 1* for 3D overview.

The online version of this article includes the following figure supplement(s) for figure 1:

**Figure supplement 1.** Identifying JONs in the EM volume.
**Figure supplement 2.** Distribution of JON synapses.

We first located the JO-C, -E, and -F neurons in the EM volume. A confocal z-stack of a driver line (R27H08-GAL4) expressing green fluorescent protein (GFP) in these JON subpopulations was registered into the EM volume (*Figure 1—figure supplement 1A,B*; *Bogovic et al., 2018*; *Hampel et al., 2015*; *Zheng et al., 2018*). We then examined the JON axon bundle where the antennal nerve enters the brain and found that the GFP had highlighted the medial region of the

bundle where the JO-C, -E, and -F neurons were previously described to reside (*Hampel et al., 2015*; *Kamikouchi et al., 2006*). Lateral to this region were the previously reconstructed JO-A and -B neurons (*Kim et al., 2020*). We reconstructed 147 JONs within the GFP-highlighted region (*Figure 1—figure supplement 1C*). 104 were completely reconstructed, including all of their pre- and post-synaptic sites. The remaining 43 JONs were reconstructed using an autosegmentation algorithm that identified the main branches, but not finer branches or synapses (*Li et al., 2019*). These latter JONs were useful for examining gross morphology, but not for determining connectivity with other neurons.

## Reconstructed JONs form a topographical map

JON topography can be defined based on the segregated organization of the different zones in the AMMC and the stereotyped projections of the JONs to discrete subareas in each zone. We therefore compared previous light-microscopy-based descriptions of these two features (*Hampel et al., 2015*; *Kamikouchi et al., 2006*) with our EM-reconstructed JONs to categorize these neurons as projecting to specific zones (*Figure 1B–G*). This not only confirmed that we had reconstructed JO-C, -E, and -F neurons (*Figure 1C,E,F*), but revealed that we had also reconstructed JO-D neurons, as well as JO-mz neurons that project to multiple zones (*Figure 1D,G*). Thus, the projections of the different reconstructed JON subpopulations form a topographical map that resembles the one obtained from light-level anatomical analysis (*Hampel et al., 2015*; *Kamikouchi et al., 2006*).

The topographical organization described above was based on the projections of the JONs. We next addressed the extent to which this organization was reflected at the synaptic level. Previous immunohistochemical studies indicated that JON presynaptic sites are broadly distributed in the posterior regions of each zone (*Kamikouchi et al., 2006*). We found that the synapses of the 104 completely reconstructed JONs were also distributed throughout the posterior regions of their respective zones (*Figure 1H–M*). Because these synapses included both pre- and post-synaptic sites, we compared their relative distributions in each zone. Examination of the total distribution of synapses did not reveal any clear difference, as the subareas of each zone had both pre- and post-synaptic sites (*Figure 1—figure supplement 2A–F*). Taken together, our results show that different JONs project in a segregated manner and form discrete zones and subareas that contain both pre- and post-synaptic sites. Thus, JONs in these subareas can interface with downstream neurons, but they are likely also subject to regulation by other neurons.

## Contributions of morphologically distinct JON types to the JO topographical map

The EM reconstructions enabled us to next systematically identify the morphologically distinct JON types within each subpopulation, and then define how these types contribute to the topographical organization of each zone. We visually inspected the reconstructed JO-C, -D, -E, and -F neurons and found that they could be categorized into different types based on morphological similarity (*Figure 2A*, *Figure 2—figure supplements 1–4*). However, the reconstructed JO-mz neurons showed no such similarity and could not be categorized (*Figure 2—figure supplement 5A,B*). As an independent categorization method, we used the NBLAST clustering algorithm that uses spatial location and neuronal morphology to calculate similarity between neurons (*Costa et al., 2016*). In agreement with our manual annotations, the algorithm clustered most of the same JONs that we had assigned as specific types (*Figure 2—figure supplement 6A,B*). In a few cases, we found disagreement between the NBLAST clustering and our manual annotations. We opted to use the neuron type categorization that was based on our manual annotations in these instances (*Figure 2A*, see Materials and methods for further explanation).

The projections of the different JON types were found to form specific subareas within each zone (*Figure 1C–F*, types shown as different color shades). These JONs were therefore named based on their zone and subarea projections and are briefly introduced below (*Figure 2A,B*). The nine reconstructed JO-C neurons form the CM, CL, and CA subareas of zone C. These neurons were categorized into three morphologically distinct types that project to these subareas (named JO-CM, -CL, and -CA neurons, *Figure 2—figure supplement 1A–C*). The nine reconstructed JO-D neurons were categorized as two different types that form the AA, BI/BO, and DP subareas (named JO-DP and -DA neurons, *Figure 2—figure supplement 2A,B*). The 62 reconstructed JO-E neurons were

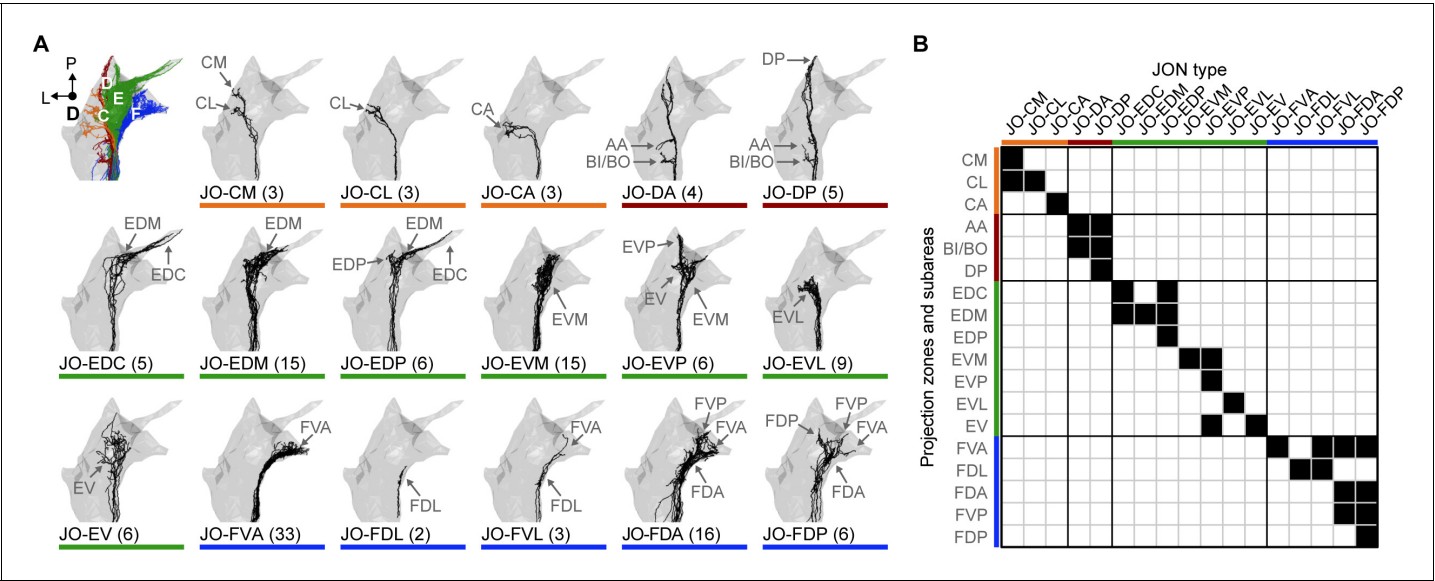

**Figure 2.** Specific JON types and their contributions to the JO topographical map. (**A**) Dorsal views of the reconstructed JONs grouped by type. For each panel, a gray mesh outlines the entire reconstructed JON population. The top left panel shows all of the reconstructed JONs colored based on their projection zones, including zones C (orange), D (red), E (green), and F (blue). The remaining panels show each JON type in black. The number of JONs shown for each type is indicated below each panel. Subareas that receive projections from each JON type are indicated with labeled arrows. Individual reconstructed JONs for each type are shown in *Figure 2—figure supplement 1*, *Figure 2—figure supplement 2*, *Figure 2—figure supplement 3*, and *Figure 2—figure supplement 4*. Note that the JO-mz neurons are not shown because they could not be categorized into types. (**B**) Grid showing the projection zones and subareas of each JON type. Zone subareas that receive projections from each JON type are indicated with black squares. Colored lines indicate the zone for each subarea and each JON type (same color scheme used in **A**).

The online version of this article includes the following figure supplement(s) for figure 2:

**Figure supplement 1.** Individual reconstructed zone C-projecting JONs.

**Figure supplement 2.** Individual reconstructed zone D-projecting JONs.

**Figure supplement 3.** Individual reconstructed zone E-projecting JONs.

**Figure supplement 4.** Individual reconstructed zone F-projecting JONs.

**Figure supplement 5.** Individual reconstructed multiple zone-projecting JONs.

**Figure supplement 6.** Correspondence between manual annotation and NBLAST clustering in the categorization of JONs.

**Figure supplement 7.** JON-to-JON synaptic connectivity.

categorized into seven types that form the EDC, EDM, EDP, EVM, EVP, EVL, and EV subareas (named JO-EDC, -EDM, -EDP, -EVM, -EVP, -EVL, and -EV neurons, *Figure 2—figure supplement 3A,B*). Lastly, the 60 reconstructed JO-F neurons were categorized into five types that form the FVA, FDA, FDP, FDL, and FVP subareas (named JO-FVA, -FDA, -FDP, -FDL, and -FVL neurons, *Figure 2—figure supplement 4A,B*). A full description of the morphologically distinct JON types, their different zone and subarea projections, and rationale for naming each type is provided in the Materials and methods. These descriptions reveal the contributions of each neuronal type to the JO topographical map (see *Video 1* for 3D overview).

## JON axons make synaptic connections with each other

Analysis of all-to-all connectivity among the different reconstructed JONs revealed that they make synaptic connections with each other. Furthermore, the connectivity tended to occur most frequently among JONs that belonged to the

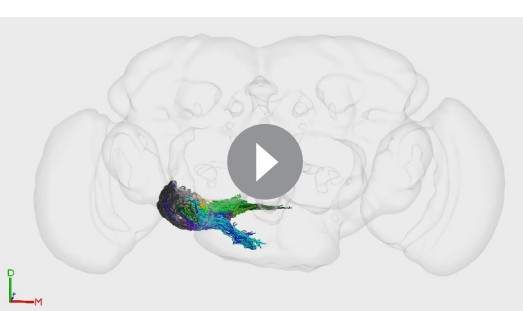

**Video 1.** EM-reconstructed JONs. Shown are the different JON types for each subpopulation.
https://elifesciences.org/articles/59976#video1

same subpopulation (*Figure 2—figure supplement 7*, *Supplementary file 2*). For example, the JO-F neurons had numerous connections with each other but showed virtually no connectivity with JO-C or -E neurons. Thus, JONs that project to the same zones show synaptic connectivity, while JONs projecting to different zones show little or no connectivity.

## Driver lines that express in JO-C and -E or -F neurons

We next produced transgenic driver lines that would enable us to compare the anatomical, physiological, and behavioral properties of the JO-C, -E, and -F neurons. New lines were necessary because the previously reported drivers that express in JO-C and -E neurons also express in other neurons. Further, although we previously described a 'clean' line that expresses in JO-F neurons (aJO-spGAL4-1) (*Hampel et al., 2015*), here we obtained additional lines to expand our toolkit for genetically accessing these JONs. We used a Split GAL4 (spGAL4) screening approach to produce four different lines that expressed in JONs whose activation could elicit antennal grooming (see Materials and methods for details, *Figure 3A–D*, *Figure 3—figure supplement 1A–D*). Two of the identified drivers express in both JO-C and -E neurons and were named JO-C/E-1 (spGAL4 combination: VT005525-AD ∩ R27H08-DBD) and JO-C/E-2 (R39H04-AD ∩ R27H08-DBD) (*Figure 3A,B*). The other two lines express mainly in JO-F neurons and were named JO-F-1 (R25F11-AD ∩ R27H08-DBD) and JO-F-2 (VT050231-AD ∩ R27H08-DBD) (*Figure 3C,D*). Our analysis of all four driver lines revealed no evidence of JO-A, -B, -D, or -mz neurons in their expression patterns.

The EM-reconstructed JONs were next used to assess which JON types were targeted by the JO-C/E-1 and −2 driver lines. The subareas formed by the reconstructed JONs (*Figure 3E*, left) were compared with those observed in the confocal light-microscopy images of each driver line expressing mCD8::GFP (*Figure 3E*, middle and right). Both lines express in JONs projecting to the subareas CL, EDP, EVP, EDM, EDC, and EVM. Because each subarea is formed by specific JON types (*Figure 2B*), we could deduce that both lines express in JO-CL, -EDC, -EDM, -EDP, -EVM, and -EVP neurons (*Figure 3G*). Importantly, neither line expresses in JO-F neurons as there are no ventral-projecting JONs in their patterns (*Figure 3A,B*). We concluded that the JO-C/E-1 and −2 driver lines express specifically in JO-C and -E neurons (*Figure 3G*).

The JO-F-1 and −2 driver lines express in JONs projecting to each zone F subarea (*Figure 3F*). Based on the EM-reconstructed JON types that form each subarea (*Figure 2B*), we predicted that both lines would express in JO-FVA, -FDL, -FVL, -FDA, and -FDP neurons (*Figure 3G*). However, it was unclear if the lines also expressed in JO-E neurons because of the possibility that these JONs were obscured in confocal images by the JO-FDA and -FDP neurons. Therefore, we used the multicolor flipout (MCFO) method (*Nern et al., 2015*) to stochastically label individual JONs within each pattern, and thereby identified the JO-FDA, -FDP, -FDL, -FVL, and -FVA neurons, with the majority of them being JO-FDA neurons (*Figure 3—figure supplement 2A–E*). A portion of the labeled JONs projected to zone E and had a posterior projection, leading us to propose they are JO-EVP neurons (*Figure 3—figure supplement 2F*). However, the lines only weakly labeled the EVP subarea as compared with the JO-F subareas (*Figure 3E,F*), suggesting that a relatively small number of JO-EVP neurons are labeled. We concluded that JO-F-1 and −2 express mostly in JO-F neurons, but also in JO-EVP neurons. Of note, JO-F-1 and −2 appear to express in the same JON types as our previously reported JO-F driver line named aJO-spGAL4-1 (*Hampel et al., 2015*).

To visualize the extent to which the JO-C/E and -F driver lines express in distinct JON subpopulations, we computationally aligned confocal stacks of their expression patterns (*Figure 3H,I*, left panels). This shows how the different driver lines express in JON subpopulations that project into distinct zones. Further, the morphology of the aligned projections was very similar to the EM-reconstructed JO-C/E and -F neurons (*Figure 3H,I*, right panels). This provides further support that the different lines selectively target the JO-C/E or -F neurons.

We next compared the distributions of the JONs that are labeled by the different driver lines in the JO chordotonal organ. The JON cell bodies are organized into a bottomless bowl-shaped array in the second antennal segment that can be visualized by labeling the JON nuclei using an antibody against the ELAV protein (*Figure 4A,B*). Expression of GFP under control of JO-C/E-1 and −2 labeled JON cell bodies in a ring around the JO bowl (*Figure 4C,D*). In contrast to the previously published JO-C/E drivers that showed expression around the entire ring of the JO bowl (*Kamikouchi et al., 2006*), JO-C/E-1 and −2 showed only sparse expression around the anterior dorsal (A-D) portion of the bowl (*Figure 4C',D'*). JO-F-1 and −2 showed expression in two clusters in

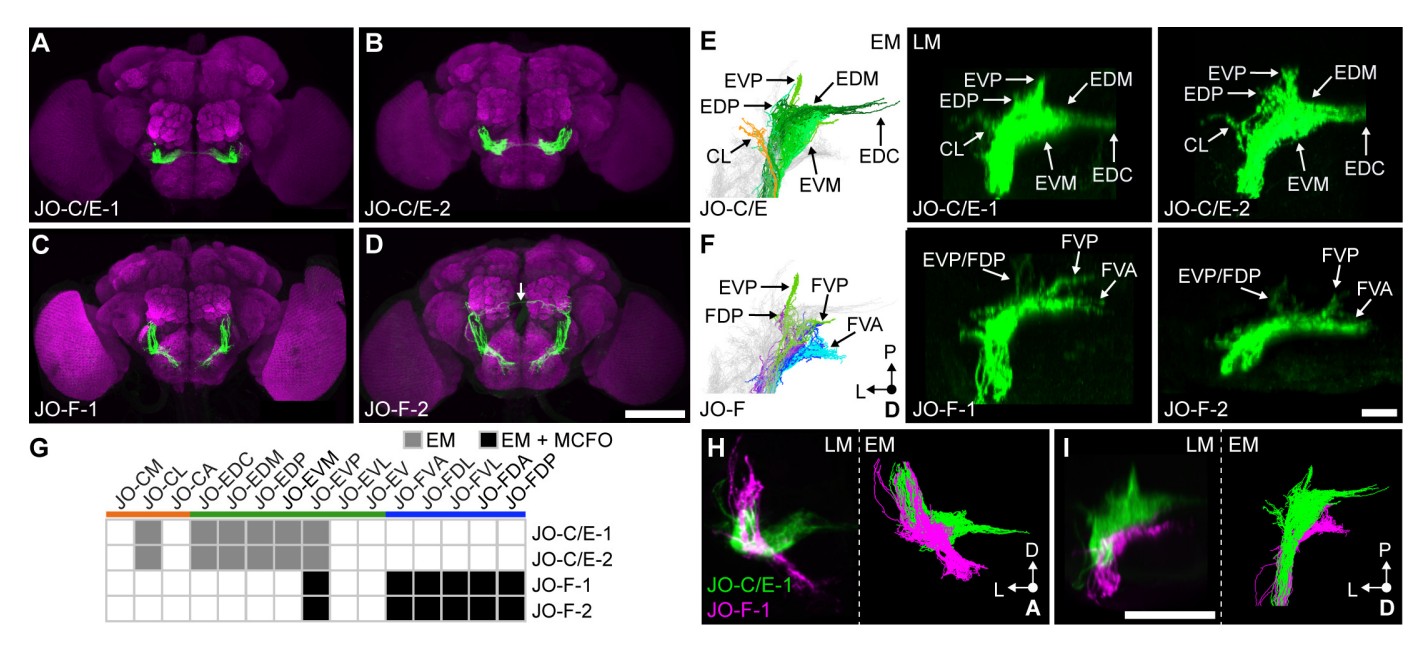

**Figure 3.** Driver lines that express in JO-C/E or JO-F neurons. (A–D) Shown are maximum intensity projections of brains (anterior view) in which driver lines JO-C/E-1 (A), JO-C/E-2 (B), JO-F-1 (C), and JO-F-2 (D) drive expression of green fluorescent protein (mCD8::GFP). Brains were immunostained for GFP (green) and Bruchpilot (magenta). The arrow shown in (D) indicates a neuron that is not a JON. Scale bar, 100 μm. *Figure 3—figure supplement 1* shows the ventral nerve cord expression pattern for each line. (E, F) Dorsal view of EM-reconstructed JON types (left panels) that are predicted to be in the expression patterns of the confocal light-microscopy (LM) images of driver-labeled neurons (middle and right panels). Driver line expression patterns of JO-C/E-1 (middle) and −2 (right) shown in (E) and JO-F-1 (middle) and −2 (right) shown in (F). Subareas are indicated with arrows. Note that in (F) the subareas FDL and FDA are not labeled because they are not visible in the dorsal view. Scale bar, 20 μm. (G) Table of JON types that are predicted to be in each driver expression pattern. The shading of each box indicates whether the predictions are supported by EM reconstructions alone (gray), or by EM and MCFO data (black). MCFO data is shown in *Figure 3—figure supplement 2*. (H, I) Computationally aligned expression patterns of JO-C/E-1 (green) and JO-F-1 (magenta) from anterior (H) and dorsal (I) views (left panels) in comparison with the EM-reconstructed JONs (right panels). Scale bar, 50 μm.

The online version of this article includes the following figure supplement(s) for figure 3:

**Figure supplement 1.** Driver lines that express in JO-C/E or -F neurons.

**Figure supplement 2.** Stochastic labeling of individual JONs in the JO-F-1 and JO-F-2 expression patterns.

the dorsal and ventral regions of the JO bowl (*Figure 4E,F*), in agreement with our previous results (*Hampel et al., 2015*). The dorsal expression was in the anterior and posterior regions of the bowl (A-D and P-D), while the ventral expression was largely restricted to the posterior (P-V) region (*Figure 4E′,F′*). In contrast to what we previously reported, JO-F-1 and −2 expression was not restricted to the dorsal and ventral clusters, but was also in more intermediate JONs in the posterior part of the bowl. This prompted us to reexamine JO-F driver lines from our previous work for evidence that they also expressed in these intermediate JONs (*Hampel et al., 2015*). Indeed, these lines show relatively faint GFP signal in intermediate JONs in the posterior bowl (not shown). This suggests that the distribution of JONs targeted by the different JO-F driver lines is more continuous, rather than restricted to clusters. A comparison of the distributions of JONs that are targeted by the JO-C/E and -F driver lines revealed that they occupy common (P-D and P-V) and distinct (A-D [JO-F] and A-V [JO-C/E]) regions of the JO bowl (*Figure 4B′–F′*).

## JO-C/E and -F neurons respond differently to mechanical stimulation of the antennae

We next compared the responses of the JO-C/E and -F neurons to mechanical stimulation of the antennae using a previously established preparation (*Matsuo et al., 2014*). Flies expressing the fluorescence-based calcium indicator GCaMP6f (*Chen et al., 2013*) in the JONs targeted by the different driver lines were immobilized and their mouthparts removed to obtain optical access to the JON

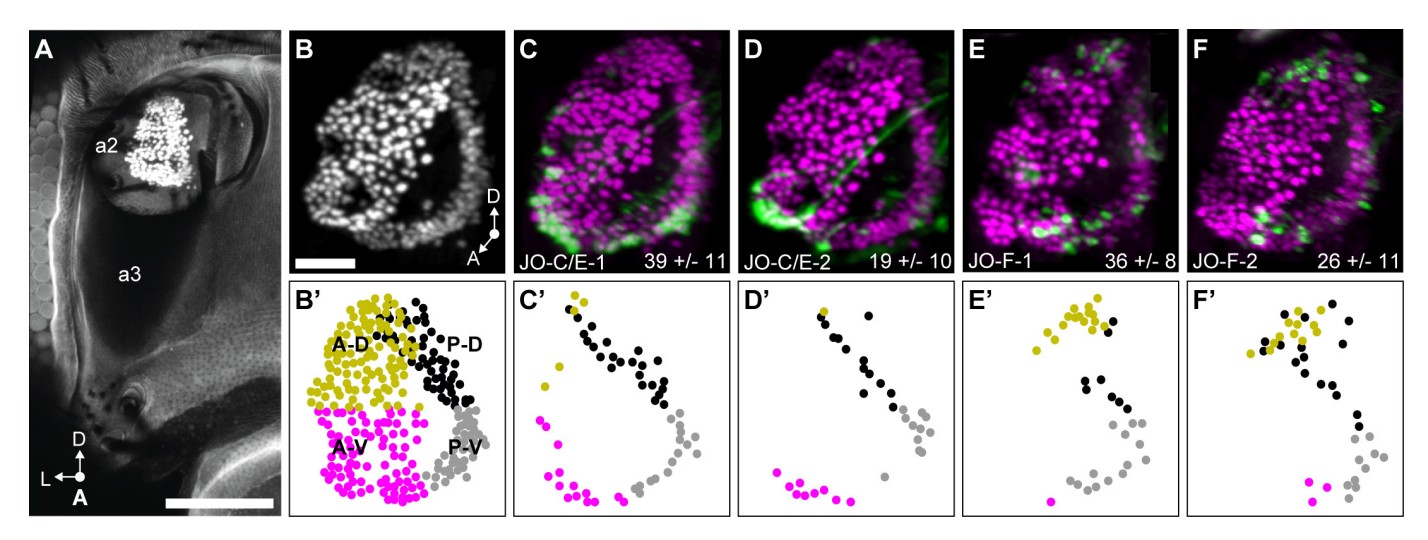

**Figure 4.** JO chordotonal organ distribution of JONs that are targeted by JO-C/E and F driver lines. (**A**) Anterior view of the antennal region of the head with the JON nuclei labeled with an anti-ELAV antibody in the second antennal segment (labeled a2, third segment is labeled a3). A maximum intensity projection is shown. The head is visualized as autofluorescence from the cuticle. Scale bar, 100 µm. (**B**) JON nuclei laterally rotated about the ventral/dorsal axis (~40°). Scale bar, 25 µm. (**C–F**) Driver lines expressing GFP in the JO. Shown is immunostaining of GFP (green) and ELAV (magenta). The average number of JONs labeled in each line ±the standard deviation is shown in the bottom right corner. (**B'**) Anterior view of manually labeled JON cell bodies in different regions from the confocal stack shown in (**B**) (not laterally rotated like in (**B–F**), ~60% JON cell bodies labeled). The JO regions are color coded, including anterior-dorsal (A-D, mustard), posterior-dorsal (P-D, black), anterior-ventral (A-V, magenta), and posterior-ventral (P-V, gray). (**C'–F'**) Manually labeled JON cell bodies (dots) that expressed GFP in a confocal z-stack of each driver line. This highlights GFP-labeled JONs in the posterior JO that are difficult to view in the maximum projections shown in (**C–F**). The colors indicate the JO region where the cell body is located. Shown are JO-C/E-1 (**C,C'**), JO-C/E-2 (**D,D'**), JO-F-1 (**E,E'**), and JO-F-2 (**F,F'**).

axon terminals in the brain. Stimuli were delivered using an electrostatically charged electrode to displace the arista and third antennal segment from their resting position (*Figure 5A*). The induced rotation of the third segment about the second segment in a particular direction or sinusoidal frequency excites the JONs. Thus, we imaged calcium responses in the JON axons in the brain while different stimuli were applied.

JO-C/E neurons were previously found to respond to sustained displacements that either push or pull the antennae towards or away from the head (*Kamikouchi et al., 2009*; *Patella and Wilson, 2018*; *Yorozu et al., 2009*). In accord with this finding, the JONs labeled by the JO-C/E-1 and −2 driver lines showed increased GCaMP6f fluorescence in response to both push and pull of the arista (*Figure 5B,C*, *Figure 5—figure supplement 1A–D*). Notably, the fluorescence increase of the JO-C/E-1-labeled neurons in response to antennal pushes was lower than that of the JO-C/E-2-labeled neurons. Although both driver lines express in the same neuron types (*Figure 3G*), these different responses may indicate that the lines express in different ratios of JO-C and -E neuron types that are known to differentially respond to antennal pushes (JO-C neurons) or pulls (JO-E neurons) (*Kamikouchi et al., 2009*; *Patella and Wilson, 2018*; *Yorozu et al., 2009*). Also in agreement with previous results using the immobilized fly preparation (*Kamikouchi et al., 2009*; *Matsuo et al., 2014*; *Yorozu et al., 2009*), we found no evidence that the JO-C/E neurons responded to vibrations (*Figure 5B,C*, *Figure 5—figure supplements 1A–D*, 200 Hz test shown).

It is unknown what stimulus excites JO-F neurons. Under the experimental conditions used here, the JONs that are labeled by JO-F-1 or −2 did not respond to push or pull movements of the antennae (*Figure 5D,E*, *Figure 5—figure supplement 2A–D*). Furthermore, we could not identify a vibration frequency that could evoke a response in these JONs, including low (40 Hz, N = 2 flies), middle (200 Hz, N = 10 flies), and high frequency vibrations (400 and 800 Hz, N = 2 flies) (*Figure 5D,E*, *Figure 5—figure supplements 2A–D*, 200 Hz test shown). We confirmed that the JONs were competent to respond to stimuli by applying KCl at the end of each experiment and observing an increased GCaMP6f signal (not shown). Thus, it remains to be determined what stimulus excites these JONs (see Discussion). However, our results indicate that JO-C/E and -F neurons do not show

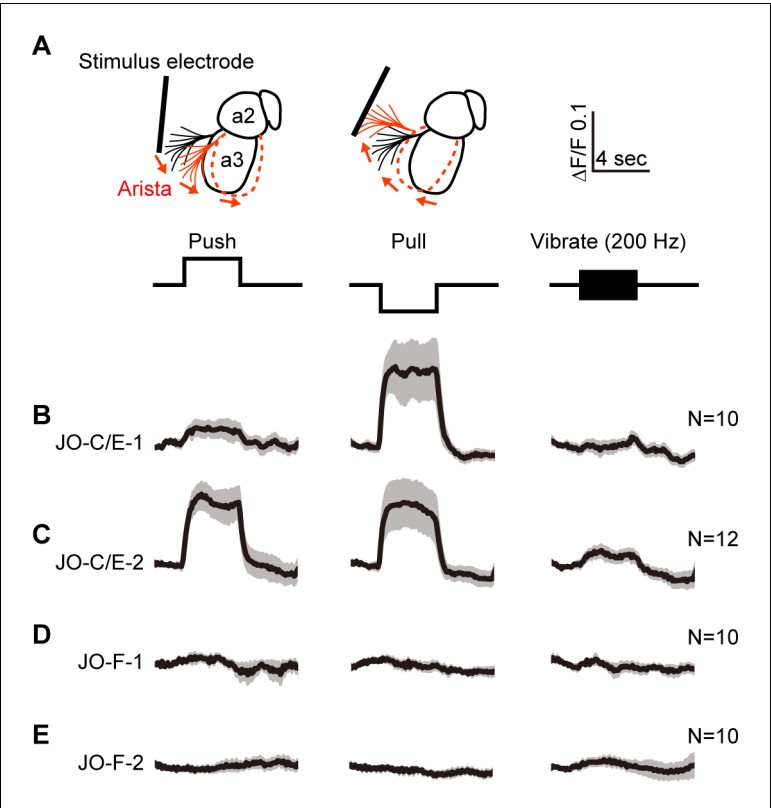

**Figure 5.** Testing the responses of JO-C/E and JO-F neurons to stimulations of the antennae. (**A**) Schematic lateral view of a fly antenna. An electrostatically charged electrode pushes or pulls the antenna via the arista towards or away from the head, respectively, or induces a 200 Hz sinusoid. (**B–E**) Calcium response of JONs to stimulations of the antennae. Flies were attached to an imaging plate, dorsal side up. The proboscis was removed to access the ventral brain for imaging GCaMP6f fluorescence changes (ΔF/F) in the JON afferents. Stimulations of the antennae were delivered for 4 s as indicated above the traces. 10 or 12 flies were tested for each driver line (N = number of flies tested). For each fly, four trials were run for each stimulus and then averaged. Each row shows the mean trace of all flies tested (black lines) from a different driver line expressing GCaMP6f, including JO-C/E-1 (**B**), JO-C/E-2 (**C**), JO-F-1 (**D**), JO-F-2 (**E**). The gray envelopes indicate the standard error of the mean. See *Figure 5—figure supplement 1* and *Figure 5—figure supplement 2* for statistical analysis.

The online version of this article includes the following figure supplement(s) for figure 5:

**Figure supplement 1.** Responses of the JO-C/E neurons to mechanical stimuli: individual traces and statistical analysis.

**Figure supplement 2.** Responses of the JO-F neurons to mechanical stimuli: individual traces and statistical analysis.

similar responses to mechanical stimuli in immobilized flies. Thus, these different JON subpopulations are both anatomically and physiologically distinct from each other.

## Activation of JO-C/E or JO-F neurons elicits common and distinct behavioral responses

We next assessed the extent to which the JO-C/E and -F neurons influence common and distinct behaviors. Our previous work implicated these subpopulations in eliciting the common behavior of antennal grooming (*Hampel et al., 2015*). In the present study, we compared the breadth of overt behavioral changes that are caused by activating either JO-C/E or -F neurons. The red light-gated neural activator CsChrimson (*Klapoetke et al., 2014*) was expressed using the different JO-C/E and -F driver lines. Flies were placed in chambers so that they could move freely and were then exposed

to red light to induce optogenetic activation of the JONs (*Hampel et al., 2017*; *Hampel et al., 2015*).

We first reproduced our previous results by showing that activation of either JO-C/E or -F neurons elicits grooming (*Figure 6A*, *Videos 2* and *3*). However, the JO-F-1 and −2 driver lines express in JO-F and -EVP neurons, which raised the possibility that the JO-EVP neurons were responsible for the grooming rather than the JO-F neurons (*Figure 3G*). To address this possibility, we identified another driver line named JO-F-3 (R60E02-LexA) that expresses exclusively in JO-F neurons and elicited grooming in the activation experiment when driving CsChrimson (*Figure 6—figure*

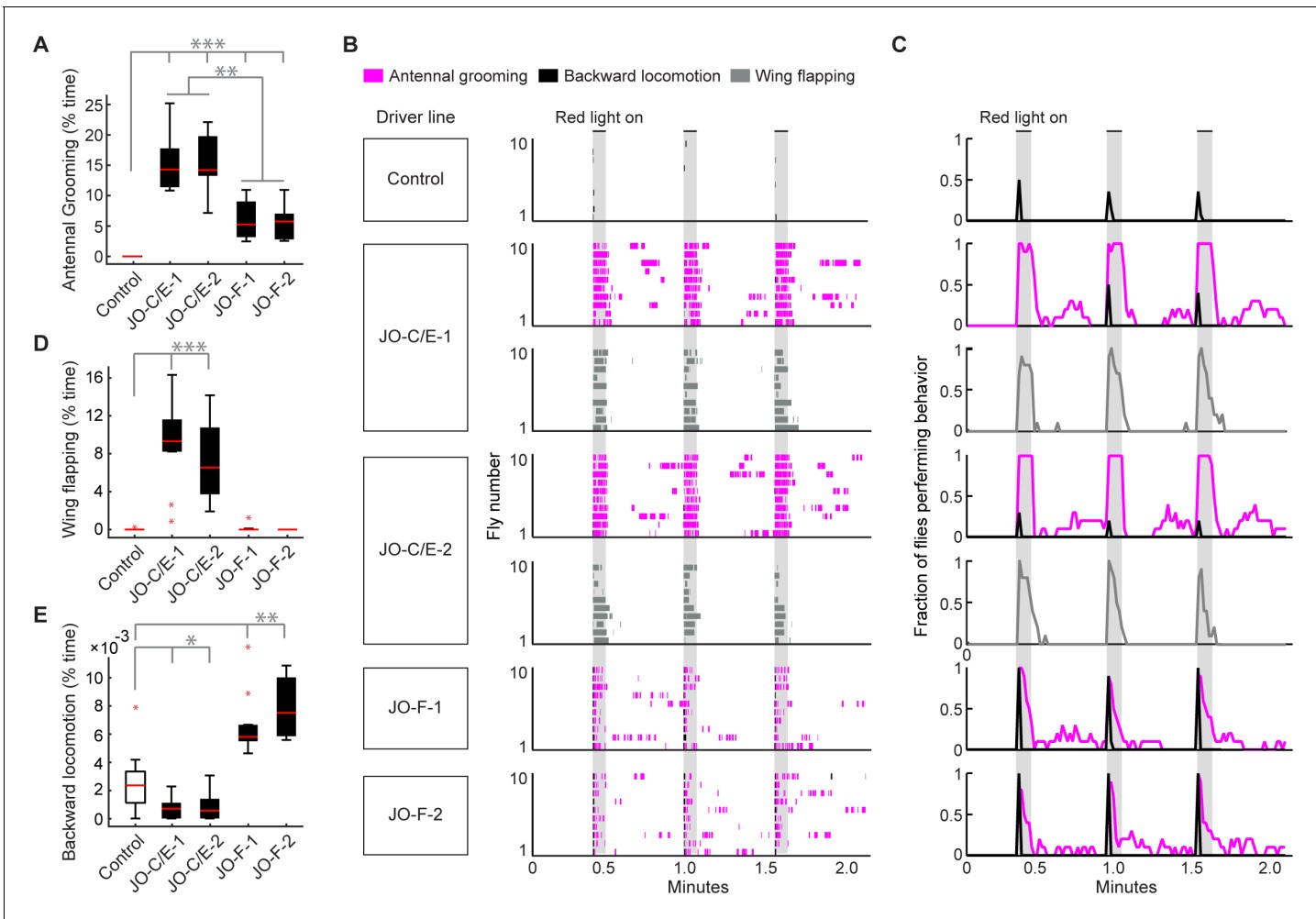

**Figure 6.** Optogenetic activation of JO-C/E or JO-F neurons elicits common and distinct behavioral responses. (A, D, E) Percent time flies spent performing antennal grooming (A), wing flapping (D), or backward locomotion (E) with optogenetic activation of JONs targeted by JO-C/E-1, JO-C/E-2, JO-F-1, and JO-F-2. Control flies do not express CsChrimson in JONs. Bottom and top of the boxes indicate the first and third quartiles respectively; median is the red line; whiskers show the upper and lower 1.5 IQR; red dots are data outliers. N ≥ 10 flies for each box; asterisks indicate *p<0.05, **p<0.001, ***p<0.0001, Kruskal–Wallis and post-hoc Mann–Whitney U pairwise tests with Bonferroni correction. *Figure 6—source data 1* contains numerical data used for producing each box plot. (B) Ethograms of manually scored videos show the behaviors elicited with red-light induced optogenetic activation. Ethograms of individual flies are stacked on top of each other. The behaviors performed are indicated in different colors, including antennal grooming (magenta), wing flapping (gray), and backward locomotion (black). Light gray bars indicate the period where a red-light stimulus was delivered (5 s). (C) Histograms show the fraction of flies that performed each behavior in one-second time bins. Note that only JO-C/E-1 and −2 elicited wing flapping, which was not mutually exclusive with grooming. Therefore, an extra row of wing flapping ethograms and histograms is shown for those lines. See *Video 2*, *Video 3*, and *Video 4* for representative examples.

The online version of this article includes the following source data and figure supplement(s) for figure 6:

**Source data 1.** Numerical data used for box plots.
**Figure supplement 1.** Driver line that expresses in JO-F neurons.

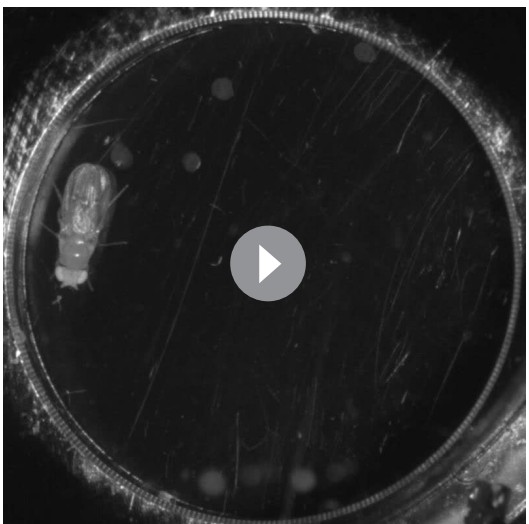

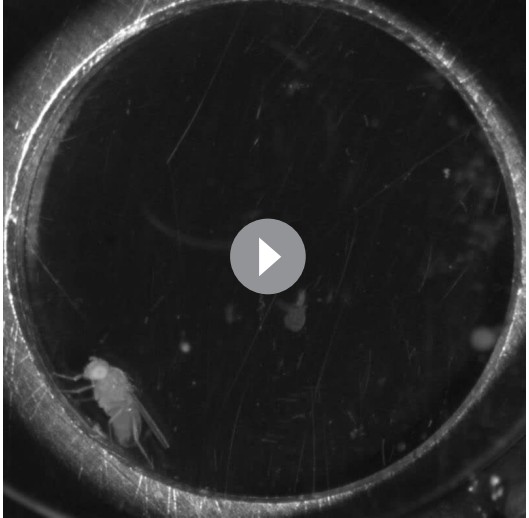

**Video 2.** Optogenetic activation of JO-C/E neurons elicits antennal grooming and wing flapping. CsChrimson was expressed in JO-C/E neurons using the JO-C/E-2 driver line. The infrared light in the bottom right corner indicates when the red light was on to activate the JO-C/E neurons.
https://elifesciences.org/articles/59976#video2

**Video 3.** Optogenetic activation of JO-F neurons elicits antennal grooming and backward locomotion. CsChrimson was expressed in JO-F neurons using the JO-F-2 driver line. The infrared light in the bottom right corner indicates when the red light was on to activate the JO-F neurons.
https://elifesciences.org/articles/59976#video3

*supplement 1A–G*). Thus, the data presented here further implicate the JO-C/E and -F neurons in antennal grooming. However, because the JO-C/E driver lines express in both JO-C and -E neurons, it remains unclear whether one or both of these subpopulations is responsible for the grooming. Addressing this will require obtaining transgenic driver lines that express exclusively in one subpopulation or the other.

This experiment further revealed that the JO-C/E and -F neurons elicit grooming that lasts for distinct durations after the onset of the red-light optogenetic stimulus (*Figure 6A–C*, *Figure 6—figure supplement 1E,F*, magenta traces). Five-second optogenetic stimulation of the JO-C/E neurons elicited grooming that lasted throughout the duration of the stimulus. In contrast, the JO-F neurons elicited shorter duration grooming that terminated prior to stimulus cessation. We considered the trivial possibility that these distinct durations of grooming were caused by differences in the number of activated JONs that were targeted in each line. However, the average number of labeled JONs did not differ markedly between the JO-C/E and -F driver lines (*Figure 4C–F*). This suggests that the distinct grooming durations were due to the physiological properties and/or functional circuit connectivity of each JON subpopulation.

In the process of annotating the grooming performed by flies with optogenetic activation of the JO-C/E or -F neurons, we observed that distinct behaviors could be elicited by each subpopulation. In the case in which we activated the

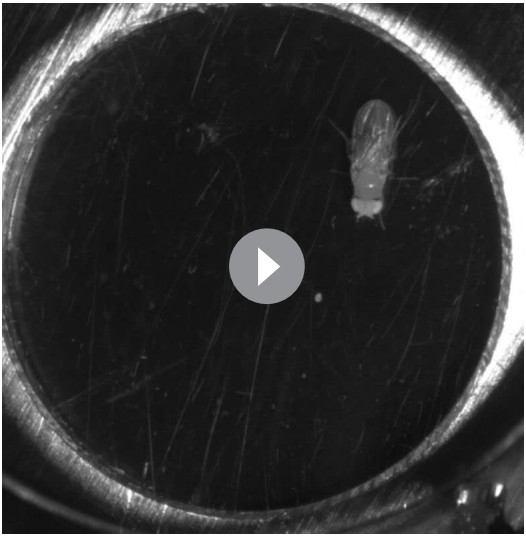

**Video 4.** Optogenetic stimulus induces backward locomotion in control flies. Control fly was exposed to the same red-light stimulus shown in *Videos 2* and *3*. The infrared light in the bottom right corner indicates when the red light was on.
https://elifesciences.org/articles/59976#video4

JO-C/E neurons, the flies were observed simultaneously grooming and performing wing flapping movements (*Figure 6B–D*, JO-C/E-1 and −2, gray trace, *Video 2*). The wings would extend to approximately 45–90-degree angles from the body axis while flapping. In contrast, activation of JO-F neurons elicited a backward locomotor response that appeared as if flies were avoiding an object that bumped into their antennae (*Video 3*, *Figure 6B,C*, *Figure 6—figure supplement 1H*). The backward locomotion and grooming were sequential and mutually exclusive, as the locomotion occurred briefly at the onset of the stimulus and was immediately followed by grooming (*Figure 6B, C*, JO-F-1 and −2, black and magenta traces). Control flies also showed backward locomotion in response to the red-light stimulus (*Figure 6B,C*, control, black trace, *Video 4*). However, less than half of these flies responded (42%), whereas nearly all of the JO-F neuron activation flies showed backward locomotion (97% for JO-F-1, 100% for JO-F-2). JO-F neuron activation also elicited longer-lasting backward locomotion than controls, with the experimental flies spending between 5- and 8-fold more time in backward locomotion than control flies (*Figure 6E*). Taken together, our results reveal that the JO-C/E and -F neurons are anatomically and physiologically distinct subpopulations that elicit both common and distinct behaviors.

## Discussion

### EM-based definition of the morphologically diverse JON subpopulations

One major goal of this work was to define the morphological diversity of the different JON subpopulations. A recent study used a serial-section EM volume of the adult fruit fly brain to reconstruct a major portion of the JO-A and -B neurons, demonstrating the utility of this approach for defining JON diversity (*Kim et al., 2020*; *Zheng et al., 2018*). In work presented here, we used this same EM volume to reconstruct the JO-C, -D, -E, -F, and -mz neurons. It remains unclear what proportion of the JO-D, -F, and -mz neurons we have reconstructed. However, it was previously estimated that there are about 200 JO-C and -E neurons total (*Kamikouchi et al., 2006*). This suggests that we reconstructed 36% of the JO-C and -E neurons (71 reconstructed out of 200).

Two lines of evidence suggested that the EM-reconstructed JONs represent the major diversity of the JO-C, -D, -E, and -F neurons. First, when the reconstructed JONs were viewed *in toto*, we could observe each previously described subarea (*Figure 1C–F*). This suggested that we had not missed JONs that are major contributors to these subareas. Second, the JONs could be categorized based on their morphological similarities to each other (*Figure 2A*, *Figure 2—figure supplements 1–4*). The fact that we reconstructed multiple morphologically similar JONs suggests that we captured the diversity of each subpopulation. Although we cannot rule out the possibility that reconstruction of more JONs would uncover additional diversity, our reconstructions provide the most comprehensive description of the JO-C, -D, -E, -F, and -mz neurons to date. In combination with the previously reconstructed JO-A and -B neurons (*Kim et al., 2020*), we now provide a near complete description of the diversity of JONs that make up each subpopulation in the JO (*Figure 1B*). This will provide a valuable resource for studies seeking to understand the neural circuit basis of the JO chordotonal organ's functions.

### Synaptic connectivity among the JONs

This study begins to address the connectivity of the JONs with the finding that they are synaptically connected with each other. This is not a new observation, as previous work showed that the JONs have axo-axonal electrical and chemical synaptic connections with each other (*Sivan-Loukianova and Eberl, 2005*). However, we find that this connectivity is largely restricted to JONs belonging to the same subpopulation, including the JO-C, -D, -E, and -F neurons (*Figure 2—figure supplement 7*). Preferential connectivity among JONs within a particular subpopulation has also been shown for the JO-A and -B neurons (*Kim et al., 2020*). This type of connectivity among sensory neurons is an emerging theme that is increasingly being described in sensory neurons across modalities (*Horne et al., 2018*; *Marin et al., 2020*; *Miroschnikow et al., 2018*; *Tobin et al., 2017*). However, the functional significance of this type of connectivity has not been addressed.

## Physiologically distinct JON subpopulations elicit grooming

In this work, we acquired new driver lines that enabled us to definitively show that the JO-C/E and -F neurons can elicit grooming of the antennae (*Figure 6A–C*). Why do these different JON subpopulations each elicit grooming? Given the evidence that the subpopulations are tuned to different mechanical stimuli (*Ishikawa et al., 2017*; *Kamikouchi et al., 2009*; *Mamiya and Dickinson, 2015*; *Matsuo et al., 2014*; *Patella and Wilson, 2018*), one possible explanation is that the JONs detect different stimuli to which the fly would appropriately respond with antennal grooming. In support of this hypothesis, different stimuli have been shown to elicit antennal grooming, including debris on the body surface (e.g. dust) and mechanical displacements of the antennae (*Hampel et al., 2015*; *Phillis et al., 1993*; *Seeds et al., 2014*). Moreover, neuronal silencing experiments have linked the JO-C/E neurons to the grooming response to dust and the JO-F neurons to the grooming response to antennal displacement: *Zhang et al., 2020* recently reported that dust-elicited grooming could be disrupted by expression of tetanus toxin using a driver line that appears to target the JO-C/E neurons, while we previously found that expression of tetanus toxin in JO-F neurons (using aJO-spGAL4-1) disrupted the grooming response to displacements of the antennae (*Hampel et al., 2015*). These studies suggest that the JON subpopulations detect these different mechanical stimuli and initiate the grooming response.

Although there is currently no physiologically based corroboration that JO-C/E neurons are tuned to respond to dust on the antennae or that JO-F neurons are tuned to antennal displacements, our present study shows that the JO-C/E and -F neurons indeed respond differently to distinct stimulations of the antennae (*Figure 5A–E*). It has been shown that the JO-C/E neurons respond to sustained pushes or pulls of the antennae (*Kamikouchi et al., 2009*; *Patella and Wilson, 2018*; *Yorozu et al., 2009*), and we confirmed that result here (*Figure 5B,C*). However, it remains to be demonstrated whether dust can sufficiently displace the antennae to excite the JO-C/E neurons. Based on our previous behavioral experiments (*Hampel et al., 2015*), we expected that sustained antennal displacements would evoke responses in the JO-F (formerly aJO) neurons. Therefore, it is puzzling that the JO-F neurons did not respond to antennal displacements here (*Figure 5D,E*). One possible limitation to the experimental approach taken in this study is that we tested the JON responses to different stimuli using an immobilized fly preparation. It is possible that a response to one of the tested stimuli could only be observed when the flies are freely behaving. In *Hampel et al., 2015*, the necessity of the JO-F neurons for the grooming response to displacements of the antennae was demonstrated using a behavioral assay whereby flies were tethered in a behavioral rig in which they were able to walk freely on a ball. There is precedent for the JONs responding differently to stimuli depending on behavioral state, as the JO-C/E neurons respond to sustained pushes and pulls of the antennae in immobilized flies, while those neurons respond to high frequency wings beats only while flies are flying (*Mamiya and Dickinson, 2015*). Therefore, responses of the JO-F neurons to antennal displacements might be observed using an experimental preparation that enables flies to move freely while being imaged.

## Neural circuit basis of JON-induced antennal grooming

How do distinct subpopulations of JONs induce antennal grooming? We previously found that the JONs elicit grooming by activating a neural circuit that elicits or 'commands' grooming of the antennae and comprises three different morphologically distinct interneuron types (*Hampel et al., 2015*). Two types are located where the JON projections terminate in the ventral brain and were named antennal grooming brain interneurons 1 and 2 (aBN1 and aBN2). The third type includes a cluster of descending neurons (aDNs) that have their dendrites in the ventral brain and axonal projections in the ventral nerve cord. The aDNs are the proposed outputs of the antennal grooming command circuit because they project to the region of the ventral nerve cord where the circuitry for generating antennal grooming leg movement patterns is presumed to be located (*Berkowitz and Laurent, 1996*; *Burrows, 1996*). The JO-C, -E, and -F neurons are in close proximity to these different interneuron types, and at least one subpopulation is functionally connected with the command circuit (*Hampel et al., 2015*). This suggests that different JON subpopulations converge onto the command circuit to control grooming behavior. The EM reconstructions established in this work provide the foundation for a future study that will address the connectivity of the JONs with the command circuit that controls antenna-directed leg movements.

## JON involvement in multiple distinct behaviors

Our study advances our understanding of the breadth of behaviors that are influenced by the JONs. The JO-C/E neurons were previously implicated in such behaviors as wind-induced suppression of locomotion, wind-guided orientation, gravitaxis, flight, and antennal grooming (*Hampel et al., 2015*; *Kamikouchi et al., 2009*; *Mamiya and Dickinson, 2015*; *Suver et al., 2019*; *Yorozu et al., 2009*). Our finding that optogenetic activation of the JO-C/E neurons results in wing flapping (*Figure 6B,C,D*) is intriguing, given that these neurons were previously shown to detect wing beats and then modulate wing movements during flight (*Mamiya and Dickinson, 2015*). Here we provided evidence that the JO-C/E neurons can also elicit wing movements. Prior to our study, the only behavior that had been ascribed to the JO-F neurons was antennal grooming (*Hampel et al., 2015*). We found that optogenetic activation of the JO-F neurons also elicits backward locomotion (*Figure 6B,C,E*). This demonstrates that, like the JO-C/E neurons, the JO-F neurons can also influence multiple distinct behaviors. Further, our results provide the first evidence that implicates locomotor avoidance as a behavior that is stimulated by the JO. Different stimuli that can move the antennae, such as unexpected mechanical displacements or static electricity have been previously shown to cause aversive locomotor responses in cockroaches (*Hunt et al., 2005*; *Jackson et al., 2011*; *Newland et al., 2008*). In stick insects, a backward locomotor response is elicited by mechanical stimulations of the antennae (*Graham and Epstein, 1985*), however, the mechanoreceptor(s) that mediate this response are unknown. Our results may suggest JO-F neurons as a link between mechanical stimulation of the antennae and backward locomotor avoidance.

Our work here reveals that the JO-C/E and -F neurons influence common and distinct behaviors, with antennal grooming as the common behavior and wing flapping and backward locomotion as the distinct behaviors. This raises the question of how these different subpopulations interface with downstream neural circuitry to control these distinct behaviors. To explain this, we hypothesize a neural circuit organization wherein the JON subpopulations have converging inputs onto the antennal grooming command circuit (discussed above) and diverging inputs onto putative circuits that control either wing flapping or backward locomotion. In the case of backward locomotion, two interneuron types (MAN and MDN) were previously identified that elicit this behavior (*Bidaye et al., 2014*). MDN was also found to be necessary for a vision-based backward locomotor response, revealing that these neurons can respond to sensory inputs (*Sen et al., 2017*; *Wu et al., 2016*). Thus, the JO-F neurons could potentially elicit an avoidance response of backward locomotion through functional connections with MAN/MDN-like neurons. The JO-C/E neurons are proposed to impinge on the wing motor system through descending circuitry (*Mamiya and Dickinson, 2015*), however the neurons in this pathway remain to be identified.

The fact that we identified multiple different JON types within each subpopulation raises the question of which types within a subpopulation (e.g. JO-EDC or -EVP neurons) control distinct behaviors. One possibility is that each type within a particular subpopulation influences a particular behavior (e.g. grooming or flight circuitry). The alternative is that all JON types within a subpopulation influence multiple distinct behaviors. Thus, the extent to which particular JON types connect with multiple different behavioral circuits or are dedicated to specific circuits remains an outstanding question.

## A resource for understanding how mechanosensory topography interfaces with neural circuits to influence behavior

The JO is a chordotonal organ in the antennae, but there are chordotonal organs in other body parts of insects and crustaceans (*Field and Matheson, 1998*). As stretch receptors, they can detect movements of particular appendages for diverse purposes, such as proprioception and sound detection. These mechanosensory structures are studied to address fundamental questions about how stimuli are processed and influence appropriate behavioral responses (*Tuthill and Wilson, 2016*). There are commonalities among chordotonal organs, as exemplified by recent studies of the fruit fly JO and leg femoral chordotonal organ (FeCO). First, subpopulations of mechanosensory neurons within these chordotonal organs are tuned to specific stimuli, such as vibrations and sustained displacements (*Kamikouchi et al., 2009*; *Mamiya et al., 2018*; *Patella and Wilson, 2018*; *Yorozu et al., 2009*). Second, these mechanosensory neurons are morphologically diverse and have topographically organized projections into the CNS (*Kamikouchi et al., 2006*; *Mamiya et al., 2018*). Third, the

subpopulations can differentially interface with downstream circuitry to influence distinct behaviors or movements (*Agrawal et al., 2020*; *Hampel et al., 2015*; *Kim et al., 2020*; *Vaughan et al., 2014*). Fourth, similar features of mechanosensory stimuli can be represented in neurons downstream of the JO and FeCO (*Agrawal et al., 2020*; *Chang et al., 2016*). These commonalities suggest that results obtained through studies of different chordotonal organs could be mutually informative. However, there is a dearth of information about how chordotonal mechanosensory neurons interface with downstream circuitry at the synaptic level. Our work, along with two recent studies (*Kim et al., 2020*; *Maniates-Selvin et al., 2020*), reveals the near complete topography of mechanosensory neurons that make up the JO and the FeCO. This provides a foundation for the rapid identification of neural circuitry that is post-synaptic to two different chordotonal organs. Ultimately, the anticipated synaptically-resolved view of the interface of the JO and FeCO with downstream circuitry will serve as a valuable resource for addressing fundamental questions about the functional significance of mechanosensory topography.

# Materials and methods

## Key resources table

| Reagent type (species) or resource | Designation | Source or reference | Identifiers | Additional information |
|---|---|---|---|---|
| Genetic reagent (*D. melanogaster*) | R27H08-GAL4 | *Jenett et al., 2012* | RRID:BDSC_49441 | |
| Genetic reagent (*D. melanogaster*) | R27H08-DBD | *Dionne et al., 2017* | RRID:BDSC_69106 | |
| Genetic reagent (*D. melanogaster*) | VT005525-AD | *Tirián and Dickson, 2017* | RRID:BDSC_72267 | aka 100C03 |
| Genetic reagent (*D. melanogaster*) | R39H04-AD | *Dionne et al., 2017* | RRID:BDSC_75734 | |
| Genetic reagent (*D. melanogaster*) | R25F11-AD | *Dionne et al., 2017* | RRID:BDSC_70623 | |
| Genetic reagent (*D. melanogaster*) | VT050231-AD | *Tirián and Dickson, 2017* | RRID:BDSC_71886 | aka 122A08 |
| Genetic reagent (*D. melanogaster*) | JO-C/E-1 | This paper | | Stock contains VT005525-AD and R27H08-DBD |
| Genetic reagent (*D. melanogaster*) | JO-C/E-2 | This paper | | Stock contains R39H04-AD and R27H08-DBD |
| Genetic reagent (*D. melanogaster*) | JO-F-1 | This paper | | Stock contains R25F11-AD and R27H08-DBD |
| Genetic reagent (*D. melanogaster*) | JO-F-2 | This paper | | Stock contains VT050231-AD and R27H08-DBD |
| Genetic reagent (*D. melanogaster*) | BPADZp; BPZpGDBD | *Hampel et al., 2015* | RRID:BDSC_79603 | spGAL4 control |
| Genetic reagent (*D. melanogaster*) | JO-F-3 (R60E06-LexA) | *Pfeiffer et al., 2010* | RRID:BDSC_54905 | |
| Genetic reagent (*D. melanogaster*) | BDPLexA | *Pfeiffer et al., 2010* | RRID:BDSC_77691 | |
| Genetic reagent (*D. melanogaster*) | 10XUAS-IVS-mCD8::GFP | *Pfeiffer et al., 2010* | RRID:BDSC_32185 | |
| Genetic reagent (*D. melanogaster*) | 20XUAS-IVS-CsChrimson-mVenus | *Klapoetke et al., 2014* | RRID:BDSC_55134 | |
| Genetic reagent (*D. melanogaster*) | 13XLexAop2-IVS-myr::GFP | | RRID:BDSC_32209 | |
| Genetic reagent (*D. melanogaster*) | MCFO-5 | *Nern et al., 2015* | RRID:BDSC_64089 | |
| Genetic reagent (*D. melanogaster*) | 20XUAS-IVS-GCaMP6f | | RRID:BDSC_42747 | |

*Continued on next page*

*Continued*

| Reagent type (species) or resource | Designation | Source or reference | Identifiers | Additional information |
|---|---|---|---|---|
| Genetic reagent (*D. melanogaster*) | *13XLexAop2-IVS-CsChrimson-mVenus* | | RRID:BDSC_55137 | |
| Antibody | anti-GFP (Rabbit polyclonal) | Thermo Fisher Scientific | Cat# A-11122, RRID:AB_221569 | IF(1:500) |
| Antibody | anti-Brp (Mouse monoclonal) | DSHB | Cat# nc82, RRID:AB_2314866 | IF(1:50) |
| Antibody | anti-ELAV (Mouse monoclonal) | DSHB | Cat# Elav-9F8A9, RRID:AB_528217 | IF(1:50) |
| Antibody | anti-ELAV (Rat monoclonal) | DSHB | Cat# Rat-Elav-7E8A10 anti-elav, RRID:AB_528218 | IF(1:50) |
| Antibody | anti-FLAG (Rat monoclonal) | Novus Biologicals | Cat# NBP1-06712, RRID:AB_1625981 | IF(1:300) |
| Antibody | anti-HA (Rabbit monoclonal) | Cell Signaling Technology | Cat# 3724, RRID:AB_1549585 | IF(1:500) |
| Antibody | anti-V5 (Mouse monoclonal) | BIO-RAD | Cat# MCA1360, RRID:AB_322378 | IF(1:300) |
| Antibody | anti-Rabbit AF488 (Goat polyclonal) | Thermo Fisher Scientific | Cat# A-11034, RRID:AB_2576217 | IF(1:500) |
| Antibody | anti-Mouse AF568 (Goat polyclonal) | Thermo Fisher Scientific | Cat# A-11031, RRID:AB_144696 | IF(1:500) |
| Antibody | anti-Rat AF568 (Goat polyclonal) | Thermo Fisher Scientific | Cat# A-11077, RRID:AB_2534121 | IF(1:500) |
| Antibody | anti-Rat AF633 (Goat polyclonal) | Thermo Fisher Scientific | Cat# A-21094, RRID:AB_2535749 | IF(1:500) |
| Chemical compound, drug | Paraformaldehyde 20% | Electron Microscopy Sciences | Cat# 15713 | |
| Chemical compound, drug | all-*trans*-Retinal | Toronto Research Chemicals | Cat# R240000 | |
| Software, algorithm | Vcode | *Hagedorn et al., 2008* | | http://social.cs.uiuc.edu/projects/vcode.html |
| Software, algorithm | Fiji | *Schindelin et al., 2012* | | http://fiji.sc/ |
| Software, algorithm | R | | | https://www.r-project.org/ |
| Software, algorithm | CMTK | *Jefferis et al., 2007* | | https://www.nitrc.org/projects/cmtk/ |
| Software, algorithm | FluoRender | *Wan et al., 2012* | | http://www.sci.utah.edu/software/fluorender.html |
| Software, algorithm | Blender version 2.79 | | | https://www.blender.org/download/releases/2-79/ |
| Software, algorithm | CATMAID | *Schneider-Mizell et al., 2016* | | https://catmaid.readthedocs.io/en/stable/ |
| Software, algorithm | MATLAB | MathWorks Inc, Natick, MA | | |
| Software, algorithm | natverse | *Bates et al., 2020* | | http://natverse.org/ |
| Software, algorithm | CATMAID-to-Blender plugin | *Schlegel et al., 2016* | | https://github.com/schlegelp/CATMAID-to-Blender |

## Rearing conditions and fly stocks

The GAL4, spGAL4, and LexA lines that were used in this study were generated by the labs of Gerald Rubin and Barry Dickson and most lines can be obtained from the Bloomington *Drosophila* stock center (*Dionne et al., 2017*; *Jenett et al., 2012*; *Pfeiffer et al., 2008*; *Tirián and Dickson, 2017*).

Control flies contain the DNA elements used for generating the different spGAL4 halves or LexA collections, but lack enhancers to drive their expression (*Pfeiffer et al., 2010*; *Pfeiffer et al., 2008*). The complete list of fly stocks that were used in this study can be found in the Key resources table.

GAL4, spGAL4, and LexA lines were crossed to their respective UAS or LexAop driver lines. Flies were reared on cornmeal and molasses food at 21°C and 50–60% relative humidity on a 16/8 hr light/dark cycle. Flies that were used for optogenetic experiments were reared on food containing 0.4 mM all-*trans*-retinal (Toronto Research Chemicals, Toronto, Canada) in vials that were wrapped in aluminum foil and covered with a box to keep them in the dark. Unless otherwise stated, flies used for experiments were male and 5 to 8 days old.

## Neural circuit reconstructions from an EM volume

Neurons and their synapses were reconstructed from a serial-section transmission electron microscopy volume of a female full adult fly brain (FAFB) at 4 × 4 × 40 nm resolution (*Zheng et al., 2018*). All reconstructions were done by an experienced tracer who used two different approaches. The first approach was based on manual annotation and provides complete reconstruction of the neurites and pre- and post-synaptic sites of each neuron. The browser-based software CATMAID (http://cat-maid.org) (*Saalfeld et al., 2009*) was used to manually navigate through the volume image stacks and manually place nodes that marked the neurites and synapses (*Schneider-Mizell et al., 2016*). For synapse annotations, we followed the criteria used by the FAFB connectomics community. Briefly, synapses had to show at least three out of the four following features: (1) an active zone with presynaptic vesicles, (2) a clear presynaptic density (such as a ribbon or T-bar), (3) a synaptic cleft, and (4) a post-synaptic density. For further tracing guidelines see *Zheng et al., 2018*. Manual tracing had the disadvantage of being labor intensive, which limited the number of neurons that could be reconstructed. Therefore, we employed the second approach of using an automated segmentation algorithm that uses flood-filling networks (*Li et al., 2019*). The algorithm would occasionally create false splits. Therefore, the tracer resolved these false splits by manually assembling the fragments as previously described (*Marin et al., 2020*). This approach enabled us to semi-automatically annotate the major branches of each neuron, but not the fine branches and synaptic sites.

To locate the JON subpopulations in the EM volume, we first registered a light-microscopy confocal z-stack of these neurons into the volume. The z-stack was of a transgenic driver line (R27H08-GAL4) that expresses in the JO-C, -E, and -F neurons (*Hampel et al., 2015*). R27H08-GAL4 (RRID: BDSC_49441) was crossed to *10XUAS-IVS-mCD8::GFP* (RRID:BDSC_32185) to label these JONs with GFP. The brains were dissected, stained, and imaged by confocal microscopy as described below (*Figure 1—figure supplement 1A*). The resulting image stack was registered into the EM volume using the software ELM (*Bogovic et al., 2018*; *Bogovic et al., 2016*) to highlight the JON axons where the antennal nerve enters the brain as a neuron bundle. The medial region of the bundle, where JO-C and -E neurons were previously described to project (*Kamikouchi et al., 2006*), was highlighted by GFP (*Figure 1—figure supplement 1B*).

We next reconstructed 147 JONs within the GFP-highlighted region (*Figure 1A,B*, *Figure 1—figure supplement 1C*, colored dots). Neuron reconstructions were performed until we had identified JO-C, -D, -E, and -F neurons, and could not uncover new morphologically distinct JONs with further reconstructions (see below for JON anatomical analysis methods). 70 JONs were manually reconstructed to completion, including all of their pre- and post-synaptic sites. We then reconstructed 77 additional JONs by assembling fragments created by the automated segmentation algorithm. 34 of these JONs were proofread using previously published methods (*Schneider-Mizell et al., 2016*), and traced to completion. Thus, out of the 147 reconstructed JONs, 104 (71%) were completely reconstructed with their entire morphology and all pre- and post-synaptic sites. At least 63% of the reconstructed JONs for each subpopulation were fully reconstructed (*Figure 2—figure supplements 1–5*, neurons marked with asterisks). All reconstructed JONs will be uploaded to a public CATMAID instance hosted by Virtual Fly Brain (https://fafb.catmaid.virtualflybrain.org/) upon publication.

Neurons were plotted and their connectivity analyzed using the natverse package (http://nat-verse.org/; *Bates et al., 2020*) in R version 3.6.2. For visualization of the AMMC neuropile, an alpha-shape was created from all nodes of the reconstructed mechanosensory neurons (147 JON skeletons in this study and the 90 JO-A and -B skeletons from *Kim et al., 2020* and transformed into a mesh object in R (alphashape3d and rgl packages)). The neurons were rendered for *Video 1* in Blender

version 2.79 with the CATMAID-to-Blender plugin (https://github.com/schlegelp/CATMAID-to-Blender; *Schlegel et al., 2016*).

## Anatomical analysis and assignment of JON types

The EM-reconstructed JONs were categorized by manual annotation and named. Assessment of the morphology of the reconstructed JONs was done using CATMAID. Annotations were done by comparing the morphology and projections of the reconstructed JONs with published light- microscopy studies (*Hampel et al., 2015*; *Kamikouchi et al., 2006*). We categorized the reconstructed JONs into 17 different types (140 JONs), and a group of 7 JO-mz neurons innervating multiple zones. See *Supplementary file 1* for detailed information on each JON type, including their FAFB skeleton ID numbers, raw and smooth cable length, number of nodes, and number of pre- and post-synaptic sites. Our rational for naming each type is provided below.

Nine of the reconstructed JONs were JO-C neurons whose projections form three different subareas (*Figure 1C*, *Figure 2A,B*). Two were previously named zone C medial (CM) and lateral (CL) (*Kamikouchi et al., 2006*). The third was a previously undescribed subarea located anterior of CM and CL (named C anterior (CA)). By examining the projections of individual reconstructed JO-C neurons, we found that each subarea is mainly formed by one of three JO-C neuron types. Two of these types project exclusively to a single subarea to form either CL or CA. However, the third type whose projections form CM has a smaller branch that projects to CL. Based on these observations, we named the three JO-C neuron types according to the subarea that receives their largest branch (named JO-CM, -CL, and -CA neurons, *Figure 2A,B*, *Figure 2—figure supplement 1A–C*).

Nine of the reconstructed JONs project to zone D (*Figure 1D*, *Figure 2A,B*). The proximal region of zone D contains protrusions that extend towards either zones A or B, and were previously named AA and BI/BO, respectively (*Kamikouchi et al., 2006*). Two different JO-D neuron types were previously described and named JO-D posterior (JO-DP) and JO-D anterior (JO-DA) neurons (*Figure 2A, B*, *Figure 2—figure supplement 2A,B*). The JO-DA neurons have branches that extend to both AA and BI/BO, and then a projection that extends towards, but does not reach the post-eriormost subarea of zone D (DP). The JO-DP neurons tend to have fewer second-order branches than JO-DA neurons, and extend a projection to DP.

62 of the reconstructed JONs project to zone E. This zone has five previously described subareas that we identified from the reconstructed JONs (*Figure 1E*, *Figure 2A,B*; *Kamikouchi et al., 2006*). The subareas are formed when JO-E neurons enter the brain and then split into two adjacent bundles called E dorsomedial (EDM) and E ventromedial (EVM). EDM curves medially and approaches the midline to form the E dorsal in the commissure (EDC) subarea. JONs that form EDC were named JO-EDC neurons, whereas most of the other JONs terminate earlier in the EDM bundle and were named JO-EDM neurons (*Figure 2—figure supplement 3A,B*). Another JON type in EDM forms a posterior protrusion called the dorsoposterior (EDP) subarea (named JO-EDP neurons). Some EDP neurons had projections that extended into the EDC subarea. JONs within the other major bundle, EVM, were divided into three types and named JO-EVM, -EVP, and -EVL neurons (*Figure 2—figure supplement 3B*). JO-EVM neurons remain in the EVM bundle. JO-EVP neurons form a protrusion from EVM that projects to the posterior brain called the E ventroposterior (EVP) subarea. JO-EVL neurons form a newly described subarea called E ventrolateral (EVL) that projects laterally from EVM, towards zone C. Some of the reconstructed JONs were not morphologically similar to the other JO-E neurons. The branches of these JONs tiled the ventralmost region of zone E (EV) and were therefore named zone E ventral (JO-EV) neurons.

60 of the reconstructed JONs project to zone F and form five subareas (*Figure 1F*, *Figure 2A,B*). The first three are formed by the proximal neurites of JO-F neurons that branch to different parts in the AMMC. We named these subareas zone F dorsoanterior (FDA), dorsoposterior (FDP), and dorsolateral (FDL). FDA is formed by JO-F neurons that extend a branch that runs adjacent to the JO-EVM neurons. Lateral and slightly ventral to FDA is the relatively small anterior protruding FDL subarea. Some JO-F neurons form the FDP subarea by extending a posterior branch that projects adjacent to the JO-EVP neurons (*Figure 1E,F*). The distal neurites of JO-F neurons project ventrally in two bundles that form the ventroanterior (FVA) and ventroposterior (FVP) subareas (*Figure 1F*). Five JO-F neuron types form the different zone F subareas (*Figure 2A,B*, *Figure 2—figure supplement 4A,B*). The first type that we named JO-FVA neurons contain few or no second-order branches and project through the AMMC and then ventrally, where most terminate their projections in the FVA subarea.

The second type that we named JO-FDA neurons project to FDA in the AMMC, and then ventrally to FVA and/or FVP. The third type that we named JO-FDP neurons project to FDA and FDP and then ventrally to the FVP subarea. The last two types that were named JO-FDL and -FVL neurons both project to the FDL subarea. These types differ in that the JO-FDL neurons terminate dorsally in the FDL subarea, whereas the JO-FVL neurons also have a ventral projection.

In contrast to the JONs that project to specific zones, we reconstructed seven JO-mz neurons that have branches projecting to multiple zones (*Figure 1G*). These JONs have been previously identified (*Kamikouchi et al., 2006*). Six out of seven reconstructed JO-mz neurons have branches that follow the posterior projections of the JO-FDP neurons, while extending projections to other zones (*Figure 1G*, *Figure 2—figure supplement 5A,B*). We could not classify the JO-mz neurons into subtypes because they showed no clear morphological similarity.

We performed an NBLAST all-to-all comparison of the 147 reconstructed JONs (*Costa et al., 2016*). We first pruned the primary axonal branch of each JON from its start point in the antennal nerve to its first branch point. Next, twigs shorter than 1 µm were pruned (*Figure 2—figure supplement 6B*, pruned neurons shown in black in the first panel). The pruning enabled us to cluster the synapse rich parts of the JONs while adjusting for any differences in neuron morphology between the manual reconstruction and automated segmentation methods. At a cut height of h = 1.4, NBLAST clustered the JONs into 15 groups that were mostly consistent with the JON types that we had identified by manual annotation (*Figure 2—figure supplement 6A,B*).

NBLAST clustered many of the same JONs that we had manually assigned as specific types, such as JO-EVL or -EVP neurons (*Figure 2—figure supplement 6A,B*, groups 3 and 14, *Supplementary file 1* shows the JON types and their NBLAST group number). NBLAST also revealed that manually assigned JON types could be further subdivided. For example, the algorithm divided JO-EVM neurons into two clusters that occupied distinct regions in the EVM subarea (*Figure 2—figure supplement 6A,B*, groups 5 and 6). In this study, we opted to keep the JO-EVM neurons as a single type as defined by our manual annotations, and based on the previously described EVM subarea boundaries (*Kamikouchi et al., 2006*). JO-mz neurons were not all clustered together, consistent with our annotation of these neurons as projecting to different zones. In some cases, NBLAST clustered JONs that we had assigned as distinct from each other. For example, some JO-FVA and -FDA neurons were clustered into the same group (*Figure 2—figure supplement 6A,B*, groups 9 and 10). These differences likely arose based on relatively small branches from the main projections of these JONs that were differentially emphasized by our manual annotations versus the NBLAST algorithm. That is, the NBLAST algorithm did not appear to emphasize JO-FDA neuron branches that formed the FDA subarea. Because the JO-FVA neurons did not project to that subarea, we opted to retain the categorization of these neuron types that was based on our manual annotations.

## Identification of driver lines that express in JON subpopulations that elicit antennal grooming

We used a spGAL4 screening approach to produce driver lines that expressed in JO-C, -E, and -F neurons. The spGAL4 system enables independent expression of the GAL4 DNA binding domain (DBD) and activation domain (AD). These domains can be reconstituted into a transcriptionally active protein when they are expressed in the overlapping cells of two different patterns (*Luan et al., 2006*; *Pfeiffer et al., 2010*). We expressed the DBD in JO-C, -E, and -F neurons using the R27H08 enhancer fragment (R27H08-DBD, RRID:BDSC_69106). To target specific subpopulations of JONs within this pattern, we identified candidate lines that were predicted to express the AD in JO-C, -E, or -F neurons (*Dionne et al., 2017*; *Tirián and Dickson, 2017*). This was done by visually screening through a database of images of the CNS expression patterns of enhancer-driven lines (*Jenett et al., 2012*). About 30 different identified candidate-ADs were crossed to flies carrying R27H08-DBD and *20XUAS-IVS-CsChrimson-mVenus* (RRID:BDSC_55134) (*Klapoetke et al., 2014*). The progeny of the different AD, DBD, and *20XUAS-IVS-CsChrimson-mVenus* combinations were placed in behavioral chambers and exposed to red light (optogenetic activation methods described below). We tested three flies for each combination, a number that we previously found could identify lines with expression in neurons whose activation elicit grooming. We then stained and imaged the CsChrimson-mVenus expression patterns of the brains and ventral nervous systems of AD/DBD combinations that elicited grooming (immunohistochemistry and imaging methods described below).

Four different DBD/AD combinations were identified that expressed in either zone C/E- or F-projecting JONs. The four AD 'hits' were VT005525-AD (RRID:BDSC_72267), R39H04-AD (RRID:BDSC_75734), R25F11-AD (RRID:BDSC_70623), and VT050231-AD (RRID:BDSC_71886).

We produced lines that contained both the AD and DBD in the same fly. Two of these lines express in JO-C and -E neurons and were named JO-C/E-1 (VT005525-AD ∩ R27H08-DBD) and JO-C/E-2 (R39H04-AD ∩ R27H08-DBD) (*Figure 3A,B*). The other two lines express mainly in JO-F neurons and were named JO-F-1 (R25F11-AD ∩ R27H08-DBD) and JO-F-2 (VT050231-AD ∩ R27H08-DBD) (*Figure 3C,D*). In a different search, we screened through the image database described above to identify a LexA driver line, R60E02-LexA (RRID:BDSC_54905), that expresses specifically in JO-F neurons (named JO-F-3). See Key resources table for more information about these driver line stock sources and references.

## Immunohistochemical analysis of the driver line expression patterns in the CNS and antennae

We evaluated the expression patterns of the different GAL4, spGAL4, and LexA driver lines using the same staining protocol. GFP or Venus-tagged CsChrimson (for spGAL4 driver line screening only) were expressed by crossing the lines to either *10XUAS-IVS-mCD8::GFP*, *20xUAS-IVS-CsChrimson-mVenus*, or *13XLexAop2-IVS-myr::GFP* (RRID:BDSC_32209). The brains, ventral nerve cords, and antennae were dissected and stained as previously described (*Hampel et al., 2015*; *Hampel et al., 2011*). The brains and ventral nerve cords were stained using anti-GFP and anti-nc82 antibodies, while the antennae were stained using anti-GFP and anti-ELAV. The following primary and secondary antibodies were used for staining: rabbit anti-GFP (Thermo Fisher Scientific, Waltham, MA, Cat# A-11122, RRID:AB_221569), mouse anti-nc82 (Developmental Studies Hybridoma Bank, University of Iowa, Cat# nc82, RRID:AB_2314866) to stain Bruchpilot, mouse anti-ELAV and rat anti-ELAV (used together for the antennal stain, Developmental Studies Hybridoma Bank, Cat# Elav-9F8A9, RRID:AB_528217 and Cat# Rat-Elav-7E8A10 anti-elav, RRID:AB_528218), goat anti-rabbit AlexaFluor-488 (Thermo Fisher Scientific, Waltham, MA, Cat# A-11034, RRID:AB_2576217), and both goat anti-mouse and goat anti-rat AlexaFluor-568 (Thermo Fisher Scientific, Cat# A-11031, RRID:AB_144696 and Cat# A-11077, RRID:AB_2534121).

For multicolor flipout (MCFO) experiments, JO-C/E-1, JO-C/E-2, JO-F-1, and JO-F-2 were crossed to the MCFO-5 stock (RRID:BDSC_64089) (*Nern et al., 2015*). 1 to 3 day old fly brains were dissected and stained using anti-V5, -FLAG, and -HA antibodies. The following primary and secondary antibodies were used: rat anti-FLAG (Novus Biologicals, LLC, Littleton, CO, Cat# NBP1-06712, RRID:AB_1625981), rabbit anti-HA (Cell Signaling Technology, Danvers, MA, Cat# 3724, RRID:AB_1549585), mouse anti-V5 (Bio-Rad, Hercules, CA, Cat# MCA1360, RRID:AB_322378), goat anti-rabbit AlexaFluor-488 (Thermo Fisher Scientific, Cat# A-11034, RRID:AB_2576217), goat anti-mouse AlexaFluor-568 (Thermo Fisher Scientific, Cat# A-11031, RRID:AB_144696), goat anti-rat AlexaFluor-633 (Thermo Fisher Scientific, Cat# A-21094, RRID:AB_2535749). We imaged individually labeled neurons from at least 10 brains for each line. Note: we made several attempts to obtain individually labeled JONs that were part of the JO-C/E-1 and −2 expression patterns. However, all of the brains that we examined showed labeling of too many neurons to visualize any one JON.

Stained CNSs and antennae were imaged using a Zeiss LSM800 confocal microscope (Carl Zeiss, Oberkochen, Germany). Image preparation and adjustment of brightness and contrast were performed with Fiji software (http://fiji.sc/). For visualizing the imaged JONs together as shown in *Figure 3H,I*, individual confocal stacks of the different spGAL4 lines were computationally aligned to the JFRC-2010 standard brain (www.virtualflybrain.org) using the Computational Morphometry Toolkit (CMTK) (https://www.nitrc.org/projects/cmtk/) (*Jefferis et al., 2007*). The aligned confocal stacks were then assembled in FluoRender (*Wan et al., 2012*; *Wan et al., 2009*), a suite of software tools for viewing image data. We compared the morphology of the JONs that were imaged via confocal microscopy with their corresponding EM-reconstructed neurons using FIJI and CATMAID, respectively.

## Testing the responses of JO-C/E and JO-F neurons to stimulations of the antennae

We tested the responses of the JON subpopulations to mechanical stimulations of the antennae using a previously published preparation (*Matsuo et al., 2014*). The JO-C/E-1, JO-C/E-2, JO-F-1, and JO-F-2 driver lines were crossed to *20XUAS-IVS-GCaMP6f* (RRID:BDSC_42747) (*Chen et al., 2013*). The progeny were cold anesthetized on ice for one minute and then attached to an imaging plate using silicon grease (SH 44M; Torray, Tokyo, Japan) with the dorsal side up. The proboscis was removed to access to the ventral brain for monitoring changes in fluorescence (*Yamada et al., 2018*). To prevent dehydration of the brain, saline solution was applied to the opening of the head. The solution contained 108 mM NaCl, 5 mM KCl, 2 mM $CaCl_2$, 8.2 mM $MgCl_2$, 4 mM $NaHCO_3$,1 mM $NaH_2PO_4$, 5 mM trehalose, 10 mM sucrose, and 5 mM HEPES, and was adjusted to pH 7.5 with 1 M NaOH, and 265 mOsm (*Wang et al., 2003*). Neural activity was monitored using a fluorescence microscope (Axio Imager.A2; Carl Zeiss, Oberkochen, Germany) equipped with a water-immersion 20x objective lens [W Achroplan/W N- Achroplan, numerical aperture (NA) 0.5; Carl Zeiss], a spinning disc confocal head CSU-W1 (Yokogawa, Tokyo, Japan), and an OBIS 488 LS laser (Coherent Technologies, Santa Clara, CA) with an excitation wavelength of 488 nm as previously described (*Yamada et al., 2018*).

Antennal displacements were induced using electrostatic forces that were generated using electrodes (*Albert et al., 2007*; *Effertz et al., 2012*; *Kamikouchi et al., 2010*; *Kamikouchi et al., 2009*). The electrical potential of the fly was increased to +15 V against ground via a charging electrode, a 0.03 mm diameter tungsten wire (Nilaco, Tokyo, Japan) that was inserted into the thorax. The following voltage commands were used: (1) sinusoids of various frequencies (200, 400, and 800 Hz), ranging from −14 V to +14 V, and (2) positive and negative steps, −50 V and +50 V for sustained push and pull displacements. These stimuli were applied for 4 s to a stimulus electrode, a 0.3 mm diameter platinum wire (Nilaco, Japan) that was placed in front of the arista of the fruit fly (*Matsuo et al., 2014*). At the end of the experiment, samples that did not show responses to any of the tested stimuli were treated with 50 μL of 4.76 M KCl that was pipetted into the saline solution (2 mL volume).

Images were acquired at a rate of 10 Hz with a 100 ms exposure time. $F_0$ was defined as the F value obtained 2.5 s before the stimulus onset. Four trials were run for each stimulus in a single fly and then averaged. 10 or 12 flies were tested for each driver line for the push, pull, and 200 Hz sinusoids. two flies were tested for the 400 and 800 Hz sinusoids. To compare the responses between 'No stimulation' (NoStim) and 'Stimulation' (Stim) conditions, we used 40 frames (4 s) before the stimulus onset for No stim and 40 frames (4 s) during the stimulus for Stim. The Wilcoxon signed-rank test was applied for the statistical analysis of the data.

## Behavioral analysis procedures

We tested for behavioral changes that are caused by activating either JO-C/E or -F neurons. JO-C/E-1, JO-C/E-2, JO-F-1, JO-F-2, and BPADZp; BPZpGDBD (spGAL4 control, RRID:BDSC_79603) were crossed to *20XUAS-CsChrimson-mVenus*. JO-F-3 and BDPLexA (LexA control, RRID:BDSC_77691) were crossed to *13XLexAop2-IVS-CsChrimson-mVenus* (RRID:BDSC_55137). The optogenetic behavioral rig, camera setup, and methods for the recording and behavioral analysis of freely moving flies were described previously (*Hampel et al., 2015*; *Seeds et al., 2014*). In brief, we used 656 nm red light at 27 mW/$cm^2$ intensity (Mightex, Toronto, Canada) for activation experiments using CsChrimson. The red-light stimulus parameters were delivered using a NIDAQ board controlled through Labview (National Instruments, Austin, TX). The red-light frequency was 5 Hz for 5 s (0.1 s on/off), and 30 s interstimulus intervals (total of 3 stimulations). Manual scoring of grooming behavior captured in prerecorded video was performed with VCode software (*Hagedorn et al., 2008*) and the data was analyzed in MATLAB (MathWorks Incorporated, Natick, MA). Antennal grooming was scored as previously described (*Hampel et al., 2015*; *Seeds et al., 2014*), however, in this work the wing flapping and backward locomotor responses are newly described. Backward locomotion was scored when the fly body moved backward by any amount. Wing flapping was scored when the wings started moving to the sides or up and down until no further movement was observed. Behavioral data was analyzed using nonparametric statistical tests as we previously reported (*Hampel et al., 2017*; *Hampel et al., 2015*). We performed a Kruskal-Wallis (ANOVA) test to compare more than

three genotypes with each other. After that we used a post-hoc Mann-Whitney U test and applied Bonferroni correction. The changes in grooming that we observed by activating the different JON subpopulations had a comparable effect size to our previously published work (*Hampel et al., 2017*; *Hampel et al., 2015*). Therefore, at least 10 experimental and 10 control flies were tested (95% power to detect a 1.48 effect size at a 0.05 significance level).

## Acknowledgements

We thank Gerald Rubin and Barry Dickson for providing split GAL4 lines; Eric Hoopfer for MATLAB code used to analyze behavioral data; Steven Sawtelle for constructing the behavioral rig and opto-genetic setup; Karen Hibbard and Todd Laverty for flies and organizational help; Tom Kazimiers for development and maintenance of CATMAID; John Bogovic for aligning light- microscopy images to the EM dataset using ELM; Zari Zavala-Ruiz and the Janelia Visiting Scientist Program for enabling our initial circuit reconstructions; Greg Jefferis and Marta Costa for hosting Katharina Eichler at Cambridge University, Maria Sosa and Bethzaida Birriel for help with visitor agreements and visas; Steven Calle-Schuler for the EM video; Gwyneth Card, Ruchi Parekh, Shada Alghailani, and Hyunsoo Kim for contributing reconstructions of JO-D neurons; and Carmen Robinett for comments on the manuscript.

## Additional information

### Funding

| Funder | Grant reference number | Author |
| --- | --- | --- |
| Whitehall Foundation | 2017-12-69 | Andrew M Seeds |
| National Institute on Minority Health and Health Disparities | MD007600 | Andrew M Seeds |
| National Institute of General Medical Sciences | GM103642 | Stefanie Hampel Andrew M Seeds |
| Puerto Rico Science, Technology and Research Trust | 2020-00195 | Andrew M Seeds |
| National Science Foundation | HRD-1736019 | Andrew M Seeds |

The funders had no role in study design, data collection and interpretation, or the decision to submit the work for publication.

### Author contributions

Stefanie Hampel, Conceptualization, Formal analysis, Supervision, Funding acquisition, Investigation, Writing - review and editing; Katharina Eichler, Software, Formal analysis, Investigation, Methodology, Writing - original draft, Writing - review and editing; Daichi Yamada, Formal analysis, Investigation; Davi D Bock, Resources, Software, Methodology; Azusa Kamikouchi, Formal analysis, Validation, Investigation, Writing - review and editing; Andrew M Seeds, Conceptualization, Formal analysis, Supervision, Funding acquisition, Investigation, Writing - original draft, Writing - review and editing

### Author ORCIDs

Stefanie Hampel (iD) https://orcid.org/0000-0001-8287-549X
Katharina Eichler (iD) http://orcid.org/0000-0002-7833-8621
Davi D Bock (iD) https://orcid.org/0000-0002-8218-7926
Azusa Kamikouchi (iD) http://orcid.org/0000-0003-1552-6892
Andrew M Seeds (iD) https://orcid.org/0000-0002-4932-6496

### Decision letter and Author response

Decision letter https://doi.org/10.7554/eLife.59976.sa1
Author response https://doi.org/10.7554/eLife.59976.sa2

# Additional files

## Supplementary files

• Supplementary file 1. Detailed information about the EM-reconstructed JONs. Includes the JON FAFB skeleton ID numbers, raw and smooth cable length, number of nodes, number of pre- and post-synaptic sites, and NBLAST group numbers.

• Supplementary file 2. JON all-to-all connectivity matrix. Shows the number of synapses for each JON-to-JON connection (presynaptic neurons – rows, post-synaptic – columns).

• Transparent reporting form

## Data availability

Neuron reconstructions will be available on https://fafb.catmaid.virtualflybrain.org/.

The following datasets were generated:

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
