## [Decision Letter]

**Acceptance summary:**

This is a Research Advance which builds upon Hampel et al., 2015. Here the authors painstakingly define the morphological subtypes of antennal Johnston's Organ mechanoreceptive neurons (JONs). They then develop driver lines to study the different populations with optogenetics and Ca-imaging. They reconfirm the activation of the subpopulations to mechanical stimuli and show that optogenetic activation of C/E and F populations elicit grooming but also other behaviors – wing flapping and avoidance for C/E and F populations respectively. The work clearly advances our knowledge of this sensory organ in flies and pushes us forward to a better understanding of how topography of sensory projections and their sub-modalities influence behavior and potentially connectivity.

**Decision letter after peer review:**

Thank you for submitting your article "Convergence of distinct subpopulations of mechanosensory neurons onto a neural circuit that elicits grooming" for consideration by *eLife*. Your article has been reviewed by three peer reviewers, including Ronald L Calabrese as the Senior Editor, Reviewing Editor and Reviewer #1.

The reviewers have discussed the reviews with one another and the Reviewing Editor has drafted this decision to help you prepare a revised submission.

Summary:

This is a Research Advance which builds upon a previous paper 'A neural command circuit for grooming movement control' (available at [https://elifesciences.org/articles/08758]). Here the authors painstakingly define the morphological subtypes of antennal Johnston's Organ mechanoreceptive neurons (JONs -particularly C, E and F populations that elicit antennal grooming). They then develop driver lines to study the different populations with optogenetics and Ca-imaging. They reconfirm the activation of the subpopulations to mechanical stimuli and show that optogenetic activation of C/E and F populations elicit grooming but also other behaviors – wing flapping and avoidance for C/E and F populations respectively. The data at this point is consistent and gives a complete description with novel interesting functional findings. The experiments are carefully done and nicely illustrated with Figures and Figure Supplements. The work clearly advances our knowledge of this sensory organ in flies and pushes us forward to a better understanding of how topography of sensory projections and their sub-modalities influence behavior and potentially connectivity.

The authors then describe their synaptic connectivity (EM reconstruction) with identified interneurons that were previously determined to be involved in eliciting antennal grooming and assess functional connectivity using optogenetic activation and Ca-imaging between the JONs and the identified downstream interneurons and among the interneurons. Here the story is less complete and there are inconsistencies between the numbers of synapses and functionally determine strength of interaction.

Essential revisions:

1) The discrepancy between the EM connectivity and the functional connectivity is substantial. The response of aBN1 to JO-F activation is very small compared to activation of JO-C/E activation but connectivity is more numerous, and JO-FDP and JO-FDA show high numbers of synapses even if across all JO-F neurons there is low average connectivity. The JO-C/E neurons give a strong functional response in aBN1, despite not that many synapses, and then indirectly to aBN2s. The arguments made (subsection “Different JON subpopulations have converging inputs onto the antennal command circuit”) are not convincing. Moreover, connections and functional connectivity with critical DNs are not explored, leaving a rather enigmatic story. This whole section should be removed from the manuscript with focus on the data of Figure 1, Figure 2, Figure 3, Figure 4, Figure 5, Figure 6 and their supplements. In future the interneurons data can be made into a more complete and more consistent advance. The title will have to be changed to reflect this revision.

2) The long descriptive subsection “The topographical organization of different JON subpopulations” should be substantially shortened. Most of this is just a list of the names they have given to different anatomical features and does not provides a lot of insight into functional organization. Could this be summarized by a table? The authors should also address the discrepancy (which appears to be a disagreement between the authors) about how many JO-C/E neurons they think there are.

(3) Another concern along the same line as #2, is in the discussion of Figure 2. After a prolonged description of "manual annotation" and NBLAST clustering of JON types, the conclusion (subsection “NBLAST clustering of the reconstructed JONs”) is that "different JON subpopulations….project to distinct zones and subareas" without telling us which description will be used in those few cases in which the two techniques disagree.

4) The JO-F neurons are not responsive to the antennal movements tested. Are these mechanoreceptors or given they are so good at evoking avoidance, are they nocioceptors?

[Editors' note: further revisions were suggested prior to acceptance, as described below.]

Thank you for resubmitting your work entitled "Distinct subpopulations of mechanosensory chordotonal organ neurons elicit grooming of the fruit fly antennae" for further consideration by *eLife*.

The manuscript has been improved but there are some remaining issues that need to be addressed before acceptance, as outlined below:

Please address the concerns of Reviewer #3.

Reviewer #3:

This is a revision of a previously submitted manuscript describing EM reconstruction of JON subtypes in the *Drosophila* antenna and the identification of additional split Gal4 lines that label subsets of these. The authors have followed the suggestions of the previous review and the new manuscript is more focused and easier to read.

My only major suggestion on the new manuscript is that the sections describing the detailed anatomy of the Split Gal4 lines (subsection “Driver lines that express in JO-C, -E, and -F neurons”) are still a bit long and don't anticipate the major functional difference between the lines shown in Figure 5B, where one C/E line responds to both directions of displacement and the other responds to only one direction. These parts of the text could be better harmonized.

---

## [Author Response]

Summary:This is a Research Advance which builds upon a previous paper 'A neural command circuit for grooming movement control' (available at [https://elifesciences.org/articles/08758]). Here the authors painstakingly define the morphological subtypes of antennal Johnston's Organ mechanoreceptive neurons (JONs -particularly C, E and F populations that elicit antennal grooming). They then develop driver lines to study the different populations with optogenetics and Ca-imaging. They reconfirm the activation of the subpopulations to mechanical stimuli and show that optogenetic activation of C/E and F populations elicit grooming but also other behaviors – wing flapping and avoidance for C/E and F populations respectively. The data at this point is consistent and gives a complete description with novel interesting functional findings. The experiments are carefully done and nicely illustrated with Figures and Figure Supplements. The work clearly advances our knowledge of this sensory organ in flies and pushes us forward to a better understanding of how topography of sensory projections and their sub-modalities influence behavior and potentially connectivity.The authors then describe their synaptic connectivity (EM reconstruction) with identified interneurons that were previously determined to be involved in eliciting antennal grooming and assess functional connectivity using optogenetic activation and Ca-imaging between the JONs and the identified downstream interneurons and among the interneurons. Here the story is less complete and there are inconsistencies between the numbers of synapses and functionally determine strength of interaction.Essential revisions:1) The discrepancy between the EM connectivity and the functional connectivity is substantial. The response of aBN1 to JO-F activation is very small compared to activation of JO-C/E activation but connectivity is more numerous, and JO-FDP and JO-FDA show high numbers of synapses even if across all JO-F neurons there is low average connectivity. The JO-C/E neurons give a strong functional response in aBN1, despite not that many synapses, and then indirectly to aBN2s. The arguments made (subsection “Different JON subpopulations have converging inputs onto the antennal command circuit”) are not convincing. Moreover, connections and functional connectivity with critical DNs are not explored, leaving a rather enigmatic story. This whole section should be removed from the manuscript with focus on the data of Figure 1, Figure 2, Figure 3, Figure 4, Figure 5, Figure 6 and their supplements. In future the interneurons data can be made into a more complete and more consistent advance. The title will have to be changed to reflect this revision.

We have removed Figure 7 and Figure 8 from the manuscript as requested by the reviewers. This required us to change different parts of the manuscript text (e.g. Title and Abstract). The main goal of these changes was to shift the emphasis of the manuscript from understanding how the Johnston’s organ neurons (JONs) interface with the antennal grooming command circuit, to defining the anatomical, physiological, and behavioral diversity of the different JON subpopulations that elicit grooming. Most of the changes are described below in relation to other reviewer comments.

2) The long descriptive subsection “The topographical organization of different JON subpopulations” should be substantially shortened. Most of this is just a list of the names they have given to different anatomical features and does not provides a lot of insight into functional organization. Could this be summarized by a table?

We respond to revision requests 2 and 3 together (see below).

(3) Another concern along the same line as #2, is in the discussion of Figure 2. After a prolonged description of "manual annotation" and NBLAST clustering of JON types, the conclusion (subsection “NBLAST clustering of the reconstructed JONs”) is that "different JON subpopulations….project to distinct zones and subareas" without telling us which description will be used in those few cases in which the two techniques disagree.

Below we respond to revision requests 2 and 3 together.

A primary reason why we expect that this manuscript will be cited is for its systematic description of a major portion of the JONs. Therefore, in our original draft we included a detailed description of the anatomy of the JONs and our logic for categorizing and naming each JON type. We also described grey areas that we encountered when naming the different JONs, such as the discrepancies between our manual assignments of the JON types and the NBLAST clustering. However, based on the reviewers’ request that we shorten our descriptions in the Results section, we see that this level detail detracted from the more generally interesting conclusions of the manuscript. We were also asked to be more definitive in our naming of the different JON types.

In order to balance the need for brevity with our expectation that the details will be important for the active community of researchers who study the JO, we have moved the above-mentioned text to a new subsection “Anatomical analysis and assignment of JON types”. This text from the previous version of the manuscript includes both the anatomical and NBLAST descriptions (subsection “Driver lines that express in JO-C, -E, and -F neurons” and subsection “JO-C/E and -F neurons respond differently to mechanical stimulation of the antennae”). We also added text to this section that briefly explains our rational for naming each JON type in light of disagreement between our manual assignments and the NBLAST results. These changes enable the more definitive and simplified description of the JON types that we now present in the Results section (described below).

The Anatomical description of the EM-reconstructed JONs in the Results is now substantially shorter and presented in a more definitive way that avoids lengthy discussion of the above-mentioned grey areas. This description is in the section entitled: “Contributions of morphologically distinct JON types to the JO topographical map.” The reviewers asked if the JON anatomy could be summarized as a table rather than described at length in the text. Inspired by this suggestion, we created a Figure that summarizes the different JON types, and creates a visual representation of the text that the reviewers requested that we remove (see new Figure 2). Figure 2A shows the morphology of each JON type, while Figure 2B indicates the zone and subarea projections of each type. We also made the NBLAST results a Figure supplement instead of a main Figure (former Figure 2 is now Figure 2—figure supplement 6). We realized that the NBLAST results were better included in the manuscript as supporting data, rather than as a main figure.

The authors should also address the discrepancy (which appears to be a disagreement between the authors) about how many JO-C/E neurons they think there are.

In the original version of the manuscript, we discussed a discrepancy in the literature about the estimated numbers of neurons in each JON subpopulation. That is, Kamikouchi et al., 2006 provided estimates of the numbers of JO-C, -D, -E, and -mz neurons based on a stochastic labeling study. From this data we estimated to have reconstructed 29% of the JO-C neurons (9 out of an estimated 31 neurons), 69% of the JO-D neurons (9 out of 13), and 87% of the JO-E neurons (62 out of 71). However, counts of JONs targeted by a transgenic driver line that expressed specifically in the JO-C and -E neurons revealed that there were twice the number of JO-C/E neurons than were estimated from the stochastic labeling study (~200 versus 102 JO-C/E neurons, also published in Kamikouchi et al., 2006). Because this driver line definitively labels ~200 JO-C/E neurons, we the authors agreed that it provides the best estimate of the number of JO-C/E neurons. Therefore, the stochastic labeling study provided a low estimate of the numbers of JO-C/E neurons. This may also suggest that the estimates of the other subpopulations (i.e. JO-D and -mz neurons) are low, and therefore the calculated percentages of reconstructed JONs for these subpopulations are overestimates. In the original manuscript, we mentioned this discrepancy in order to provide the reader with an estimate range of the numbers of JONs that we had reconstructed for each subpopulation. Based on the reviewers’ comments, we see that this caused confusion and was counterproductive.

A simplified estimate of the numbers of reconstructed JONs is now included in a new Discussion section of the revised manuscript. Given that the JO-C/E driver line mentioned above definitively labels ~200 neurons, we now use this number to estimate that we have reconstructed 36% of the JO-C/E neurons (71 reconstructed out of 200). Because the above-mentioned discrepancy casts doubt on the estimated numbers of JO-D and -mz neurons, the revised manuscript no longer uses this data to estimate the proportion of these JONs that were reconstructed.

We note that the manuscript still provides other lines of evidence to suggest that the EM-reconstructed JONs represent the major diversity of the JON subpopulations. That is, we observe all of the previously described zones and subareas with our reconstructed JONs, and that we reconstructed multiple JONs for each type. Given that reviewer #1 requested a clearer distinction between the Results section and Discussion section in the revised manuscript (see below), all of this text has been moved to the subsection “EM-based definition of the morphologically diverse JON subpopulations”.

4) The JO-F neurons are not responsive to the antennal movements tested. Are these mechanoreceptors or given they are so good at evoking avoidance, are they nocioceptors?

This is a possible and interesting hypothesis, however most of our current evidence still points to a role for the JO-F neurons as being involved in detecting displacements of the antennae. We have expanded on this hypothesis in subsection “Physiologically distinct JON subpopulations elicit grooming” and subsection “JON involvement in multiple distinct behaviors”. In the first, we describe the behavioral evidence demonstrating that the JO-F neurons are necessary for detecting displacements of the antennae, and then we suggest why the present study did not show that these JONs are physiologically tuned to such displacements. In the second of these, we describe experiments from other insects showing that displacements of the antennae can induce avoidance responses such as backward locomotion. We hope that these sections provide a clearer description of our current hypotheses about the physiological tuning of the JO-F neurons and their roles in inducing different behavioral responses, such as grooming and backward locomotor avoidance.

This reviewer comment also made us realize that our use of the term avoidance may be an over interpretation and possibly misleading, as we have not demonstrated that the backward locomotion is actually an avoidance response. Therefore, we have changed our wording that describes this to simply reflect what we observe with activation of the JO-F neurons, which is backward locomotion. We have change text in the Results section, Figure 6 and its legend, and Video 3 and Video 4 legends to reflect this change.

[Editors' note: further revisions were suggested prior to acceptance, as described below.]

Reviewer #3:This is a revision of a previously submitted manuscript describing EM reconstruction of JON subtypes in the *Drosophila* antenna and the identification of additional split Gal4 lines that label subsets of these. The authors have followed the suggestions of the previous review and the new manuscript is more focused and easier to read.My only major suggestion on the new manuscript is that the sections describing the detailed anatomy of the Split Gal4 lines (subsection “Driver lines that express in JO-C, -E, and -F neurons”) are still a bit long and don't anticipate the major functional difference between the lines shown in Figure 5B, where one C/E line responds to both directions of displacement and the other responds to only one direction. These parts of the text could be better harmonized.

The two paragraphs mentioned by the reviewer have been reduced by 19% to bring their total word count from 433 to 351. We removed two sentences from the first of the two paragraphs. The first sentence reads: “In contrast, we could not identify the CM, CA, EVL, or EV subareas in the expression patterns of either line, suggesting that JO-C/E-1 and -2 do not express in JO-CM, -CA, -EVL, and -EV neurons.” This could be inferred from Figure 3. We also removed a sentence that reads: “We sought to verify the different JON types in each driver expression pattern using a method to stochastically label individual JONs (Nern et al., 2015), but we were unable to label individual JONs for JO-C/E-1 or -2 using this method.” This sentence was not critical for the paragraph, and is also described in subsection “Immunohistochemical analysis of the driver line expression patterns in the CNS and antennae”. In the second paragraph, we removed the list of subareas that are targeted by JONs in the JO-F-1 and -2 expression patterns. The removed text reads: “including FDA, FDP, FDL, FVA, and FVP”. The authors believe that all of the remaining text is important for the description of the driver lines, and/or is necessary for understanding the later sections of the manuscript.

In the previous version of the manuscript, we did not address that there were differences between the JONs targeted by the JO-C/E-1 and -2 driver lines in their responses to pushes on the antennae. This was partially because the JONs targeted by the JO-C/E-1 and -2 driver lines both showed significant responses to pushes (Figure 5—figure supplement 1). However, as the reviewer rightfully indicates, the JO-C/E-1 response was not as strong as the one observed when we imaged the JONs that are targeted by the JO-C/E-2 driver line. The reviewer suggested that we link the description of Figure 3 with the experimental results shown in Figure 5, to presumably link our description of the expression pattern of JO-C/E-1 and -2 with their observed response difference. We suspect that the observed response difference is due to a difference in the relative ratios of JO-C and -E neurons in each pattern, given that the JO-C neurons are implicated in responding antennal pulls and the JO-E neurons respond to pushes. However, we were unable to explain this result based on our assessment of the driver line expression patterns, as we cannot determine the relative numbers of JO-C and -E neurons that are targeted by each line. Further, it is unknown whether all JO-E neuron types respond to pushes, or whether only specific JO-E neuron types respond. Therefore, the differences between the lines could be due to the relative ratios of JO-E neuron types that are labeled by each driver line. As a result, we have not been able to identify a clear-cut reason that for the differences observed in our imaging experiments.

We have addressed the reviewer comment by inserting two sentences in subsection “JO-C/E and -F neurons respond differently to mechanical stimulation of the antennae”. These sentences read: “Notably, the fluorescence increase of the JO-C/E-1-labeled neurons in response to antennal pushes was lower than that of the JO-C/E-2-labeled neurons. Although both driver lines express in the same neuron types (Figure 3G), these different responses may indicate that the lines express in different ratios of JO-C and -E neuron types that are known to differentially respond to antennal pushes (JO-C neurons) or pulls (JO-E neurons) (Kamikouchi et al., 2009; Patella and Wilson, 2018; Yorozu et al., 2009).” We think that this change better links the data shown in Figure 3 with the Figure 5 experimental result.

In the course of addressing this reviewer suggestion and reviewing the data related to Figure 5, we noticed two errors in Figure 5—figure supplement 2D. First, the Y-axis was labelled as “∆F/F 0.1”, where it should have been labeled “∆F/F”. Second, asterisks indicating p < 0.001 significance (**) were mistakenly placed in the graph for the 200 Hz stimulus, where no significant response (n.s.) should have been indicated. We have made the appropriate changes in the revised figure supplement. We apologize for these errors.